# ROBUST MODEL BASED REINFORCEMENT LEARNING USING $\mathcal{L}_1$ ADAPTIVE CONTROL

**Minjun Sung,** **Sambhu H. Karumanchi,** **Aditya Gahlawat,  Naira Hovakimyan**
Department of Mechanical Science & Engineering
University of Illinois Urbana-Champaign
Urbana, IL, 61801, USA
{mjsung2,shk9,gahlawat,nhovakim}@illinois.edu

## ABSTRACT

We introduce $\mathcal{L}_1$-MBRL, a control-theoretic augmentation scheme for Model-Based Reinforcement Learning (MBRL) algorithms. Unlike model-free approaches, MBRL algorithms learn a model of the transition function using data and use it to design a control input. Our approach generates a series of approximate control-affine models of the learned transition function according to the proposed *switching law*. Using the approximate model, control input produced by the underlying MBRL is perturbed by the $\mathcal{L}_1$ adaptive control, which is designed to enhance the robustness of the system against uncertainties. Importantly, this approach is agnostic to the choice of MBRL algorithm, enabling the use of the scheme with various MBRL algorithms. MBRL algorithms with $\mathcal{L}_1$ augmentation exhibit enhanced performance and sample efficiency across multiple MuJoCo environments, outperforming the original MBRL algorithms, both with and without system noise.

## 1 INTRODUCTION

Reinforcement learning (RL) combines stochastic optimal control with data-driven learning. Recent progress in deep neural networks (NNs) has enabled RL algorithms to make decisions in complex and dynamic environments (Wang et al., 2022). Reinforcement learning algorithms have achieved remarkable performance in a wide range of applications, including robotics (Ibarz et al., 2021; Nguyen & La, 2019), natural language processing (Wang et al., 2018; Wu et al., 2018), autonomous driving (Milz et al., 2018; Li et al., 2020), and computer vision (Yun et al., 2017; Wu et al., 2017).

There are two main approaches to reinforcement learning: Model-Free RL (MFRL) and MBRL. MFRL algorithms directly learn a policy to maximize cumulative reward from data, while MBRL algorithms first learn a model of the transition function and then use it to obtain optimal policies (Moerland et al., 2023). While MFRL algorithms have demonstrated impressive asymptotic performance, they often suffer from poor sample complexity (Mnih et al., 2015; Lillicrap et al., 2016; Schulman et al., 2017). On the other hand, MBRL algorithms offer superior sample complexity and are agnostic to tasks or rewards (Kocijan et al., 2004; Deisenroth et al., 2013). While MBRL algorithms have traditionally lagged behind MFRL algorithms in terms of asymptotic performance, recent approaches, such as the one presented by Chua et al. (2018), aim to bridge this gap.

In MBRL, learning a model of the transition function can introduce model (or epistemic) uncertainties due to the lack of sufficient data or data with insufficient information. Moreover, real-world systems are also subject to inherently random aleatoric uncertainties. As a result, unless sufficient data—both in quantity and quality—is available, the learned policies will exhibit a gap between expected and actual performance, commonly referred to as the sim-to-real gap (Zhao et al., 2020).

The field of robust and adaptive control theory has a rich history and was born out of a need to design a controller to address the uncertainties discussed above. Given that both MBRL algorithms and classical control tools depend on models of the transition function, it is natural to consider exploring the consolidation of robust and adaptive control with MBRL. However, such a consolidation is far

---
*The authors equally contributed to this work.

from straightforward, primarily due to the difference between the class of models for which the robustness is considered. To analyze systems and design controllers for such systems, conventional control methods often assume extensively modeled dynamics that are gain scheduled, linear, control affine, and/or true up to parametric uncertainties (Neal et al., 2004; Nichols et al., 1993). On the other hand, MBRL algorithms frequently update highly nonlinear models (e.g. NNs) to enhance their predictive accuracy. The combination of this iterative updating and the model's high nonlinearity creates a unique challenge in embedding robust and adaptive controllers within MBRL algorithms.

## 1.1 STATEMENT OF CONTRIBUTIONS

We propose the $\mathcal{L}_1$-MBRL framework as an add-on scheme to augment MBRL algorithms, which offers improved robustness against epistemic and aleatoric uncertainties. We *affinize* the learned model in the control space according to the *switching law* to construct a control-affine model based on which the $\mathcal{L}_1$ control input is designed. The *switching law* design provides a distinct capability to explicitly control the predictive performance bound of the *state predictor* within the $\mathcal{L}_1$ adaptive control architecture while harnessing the robustness advantages offered by the $\mathcal{L}_1$ adaptive control. The $\mathcal{L}_1$ add-on does not require any modifications to the underlying MBRL algorithm, making it agnostic to the choice of the baseline MBRL algorithm. To evaluate the effectiveness of the $\mathcal{L}_1$-MBRL scheme, we conduct extensive numerical simulations using two baseline MBRL algorithms across multiple environments, including scenarios with action or observation noise. The results unequivocally demonstrate that the $\mathcal{L}_1$-MBRL scheme enhances the performance of the underlying MBRL algorithms without any redesign or retuning of the $\mathcal{L}_1$ controller from one scenario to another.

## 1.2 RELATED WORK

**Control Augmentation of RL policies** is of significant relevance to our research. Notable recent studies in this area, including Cheng et al. (2022) and Arevalo-Castiblanco et al. (2021), have investigated the augmentation of adaptive controllers to policies learned through MFRL algorithms. However, these approaches are limited by their assumption of known nominal models and their restriction to control-affine or nonlinear models with known basis functions, which restricts their applicability to specific system types. In contrast, our approach does not assume any specific structure or knowledge of the nominal dynamics. We instead provide a general framework to augment an $\mathcal{L}_1$ adaptive controller to the learned policy, while simultaneously learning the transition function.

**Robust and adversarial RL methods** aim to enhance the robustness of RL policies by utilizing minimax optimization with adversarial perturbation, as seen in various studies (Tobin et al., 2017; Peng et al., 2018; Loquercio et al., 2019; Pinto et al., 2017). However, existing methods often involve modifications to data or dynamics in order to handle worst-case scenarios, leading to poor general performance. In contrast, our method offers a distinct advantage by enhancing robustness without perturbing the underlying MBRL algorithm. This allows us to improve the robustness of the baseline algorithm without sacrificing its general performance.

**Meta-(MB)RL** methods train models across multiple tasks to facilitate rapid adaptation to dynamic variations as proposed in (Finn et al., 2017; Nagabandi et al., 2019a;b). These approaches employ hierarchical latent variable models to handle non-stationary environments. However, they lack explicit provisions for uncertainty estimation or rejection, which can result in significant performance degradation when faced with uncertain conditions (Chen et al., 2021). In contrast, the $\mathcal{L}_1$-MBRL framework is purposefully designed to address this limitation through uncertainty estimation and explicit rejection. Importantly, our $\mathcal{L}_1$-MBRL method offers the potential for effective integration with meta-RL approaches, allowing for the joint leveraging of both methods to achieve both environment adaptation and uncertainty rejection in a non-stationary and noisy environment.

## 2 PRELIMINARIES

In this section we provide a brief overview of the two main components of our $\mathcal{L}_1$-MBRL framework: MBRL and $\mathcal{L}_1$ adaptive control methodology.

## 2.1 MODEL BASED REINFORCEMENT LEARNING

This paper assumes a discrete-time finite-horizon Markov Decision Process (MDP), defined by the tuple $\mathcal{M} = (\mathcal{X}, \mathcal{U}, f, r, \rho_0, \gamma, H)$. Here $\mathcal{X} \subset \mathbb{R}^n$ is the compact state space, $\mathcal{U} \subset \mathbb{R}^m$ is the compact action space, $f : \mathcal{X} \times \mathcal{U} \to \mathcal{X}$ is the deterministic transition function, $r : \mathcal{X} \times \mathcal{A} \to \mathbb{R}$ is a bounded reward function. Let $\xi(\mathcal{X})$ be the set of probability distributions over $\mathcal{X}$ and $\rho_0 \in \xi(\mathcal{X})$ be the initial state distribution. $\gamma$ is the discount factor and $H \in \mathbb{N}$ is a known horizon of the problem. For any time step $t < H$, if $x_t \notin \mathcal{X}$ or $u_t \notin \mathcal{U}$, then the episode is considered to have failed such that $r(x_{t'}, u_{t'}) = 0$ for all $t' = t, t+1, \ldots, H$. A policy is denoted by $\pi$ and is specified as $\pi := [\pi_1, \ldots, \pi_{H-1}]$, where $\pi_t : \mathcal{X} \to \xi(\mathcal{A})$ and $\xi(\mathcal{A})$ is the set of probability distributions over $\mathcal{A}$. The goal of RL is to find a policy that maximizes the expected sum of the reward along a trajectory $\tau := (x_0, u_0, \cdots, x_{H-1}, u_{H-1}, x_H)$, or formally, to maximize $J(\pi) = \mathbb{E}_{x_0 \sim \rho_0, u_t \sim \pi_t}[\sum_{t=1}^H \gamma^t r(x_t, u_t)]$, where $x_{t+1} - x_t = f(x_t, u_t)$ (Nagabandi et al., 2018). The trained model can be utilized in various ways to obtain the policy, as detailed in Sec. 3.1.

## 2.2 $\mathcal{L}_1$ ADAPTIVE CONTROL

The $\mathcal{L}_1$ adaptive control theory provides a framework to counter the uncertainties with guaranteed transient and steady-state performance, alongside robustness margins (Hovakimyan & Cao, 2010). The performance and reliability of the $\mathcal{L}_1$ adaptive control has been extensively tested on systems including robotic platforms (Cheng et al., 2022; Pravitra et al., 2020; Wu et al., 2022), NASA AirSTAR sub-scale aircraft (Gregory et al., 2009; 2010), and Learjet (Ackerman et al., 2017). While we give a brief description of the $\mathcal{L}_1$ adaptive control in this subsection, we refer the interested reader to Appendix A for detailed explanation.

Assume that the continuous-time dynamics of a system can be represented as

$$\dot{x}(t) = g(x(t)) + h(x(t))u(t) + d(t, x(t), u(t)), \quad x(0) = x_0, \tag{1}$$

where $x(t) \in \mathbb{R}^n$ is the system state, $u(t) \in \mathbb{R}^m$ is the control input, $g : \mathbb{R}^n \to \mathbb{R}^n$ and $h : \mathbb{R}^n \to \mathbb{R}^{n \times m}$ are known nonlinear functions, and $d(t, x(t), u(t)) \in \mathbb{R}^n$ represents the unknown residual containing both the model uncertainties and the disturbances affecting the system.

Consider a desired control input $u^\star(t) \in \mathbb{R}^m$ and the induced desired state trajectory $x^\star(t) \in \mathbb{R}^n$ based on the nominal (uncertainty-free) dynamics

$$\dot{x}^\star(t) = g(x^\star(t)) + h(x^\star(t))u^\star(t), \quad x^\star(0) = x_0. \tag{2}$$

If we directly apply $u^\star(t)$ to the true system in Equation (1), the presence of the uncertainty $d(t, x(t), u(t))$ can cause the actual trajectory to diverge unquantifiably from the nominal trajectory. The $\mathcal{L}_1$ adaptive controller computes an additive control input $u_a(t)$ to ensure that the augmented input $u(t) = u^\star(t) + u_a(t)$ keeps the actual trajectory $x(t)$ bounded around the nominal trajectory $x^\star(t)$ in a quantifiable and uniform manner.

The $\mathcal{L}_1$ adaptive controller has three components: the state predictor, the adaptive law, and a low-pass filter. The state predictor is given by

$$\dot{\hat{x}}(t) = g(x(t)) + h(x(t))(u^\star(t) + u_a(t)) + \hat{\sigma}(t) + A_s \tilde{x}(t), \tag{3}$$

with the initial condition $\hat{x}(0) = \hat{x}_0$, where $\hat{x}(t) \in \mathbb{R}^n$ is the state of the predictor, $\hat{\sigma}(t)$ is the estimate of $d(t, x(t), u(t))$, $\tilde{x}(t) = \hat{x}(t) - x(t)$ is the state prediction error, and $A_s \in \mathbb{R}^{n \times n}$ is a Hurwitz matrix chosen by the user. Furthermore, $\hat{\sigma}(t)$ can be decomposed as

$$\hat{\sigma}(t) = h(x(t))\hat{\sigma}_m(t) + h^\perp(x(t))\hat{\sigma}_{um}(t), \tag{4}$$

where $\hat{\sigma}_m(t)$ and $\hat{\sigma}_{um}(t)$ are the estimates of the *matched* and *unmatched* uncertainties. Here, $h^\perp(x(t)) \in \mathbb{R}^{n \times (n-m)}$ is a matrix satisfying $h(x(t))^\top h^\perp(x(t)) = 0$ and $\text{rank}\left(\left[h(x(t)), \, h^\perp(x(t))\right]\right) = n$. The existence of $h^\perp(x(t))$ is guaranteed, given $h(x(t))$ is a full-rank matrix. The role of the predictor is to produce the state estimate $\hat{x}(t)$ induced by the uncertainty estimate $\hat{\sigma}(t)$.

The uncertainty estimate is updated using the piecewise constant adaptive law given by

$$\hat{\sigma}(t) = \hat{\sigma}(iT_s) = -\Phi^{-1}(T_s)\mu(iT_s), \quad t \in [iT_s, (i+1)T_s), \quad i \in \mathbb{N}, \quad \hat{\sigma}(0) = \hat{\sigma}_0, \tag{5}$$

where $\Phi(T_s) = A_s^{-1}(\exp(A_s T_s) - \mathbb{I}_n)$, $\mu(iT_s) = \exp(A_s T_s)\tilde{x}(iT_s)$, and $T_s$ is the sampling time.

Finally, the control input is given by

$$u(t) = u^\star(t) + u_a(t), \quad u_a(s) = -C(s)\mathfrak{L}[\hat{\sigma}_m(t)], \tag{6}$$

where $\mathfrak{L}[\cdot]$ denotes the Laplace transform, and the $\mathcal{L}_1$ input $u_a(t)$ is the output of the low-pass filter $C(s)$ in response to the estimate of the matched component $\hat{\sigma}_m(t)$. The bandwidth of the low-pass filter is chosen to satisfy the small-gain stability condition (Wang & Hovakimyan, 2011).

## 3 THE $\mathcal{L}_1$-MBRL ALGORITHM

In this section, we present the $\mathcal{L}_1$-MBRL algorithm, illustrated in Fig. 1. We first explain a standard MBRL algorithm and describe our method to integrate $\mathcal{L}_1$ adaptive control with it.

### 3.1 STANDARD MBRL

As our work aims to develop an add-on module that enhances the robustness of an existing MBRL algorithm, we provide a high-level overview of a standard MBRL algorithm and its popular complementary techniques.

The standard structure of MBRL algorithms involves the following steps: data collection, model updating, and policy updating using the updated model. To reduce model bias, many recent results consider an ensemble of models $\{\hat{f}_{\theta_i}\}_{i=1,2,\cdots,M}$, $M \in \mathbb{N}$, on the data set $\mathcal{D}$. The ensemble model $\hat{f}_{\theta_i}$ is trained by minimizing the following loss function for each $\theta_i$ (Nagabandi et al., 2018):

$$\frac{1}{|\mathcal{D}|} \sum_{(x_t, u_t, x_{t+1}) \in D} \|(x_{t+1} - x_t) - \hat{f}_{\theta_i}(x_t, u_t)\|_2^2. \tag{7}$$

In this paper, we consider (7) as the loss func-

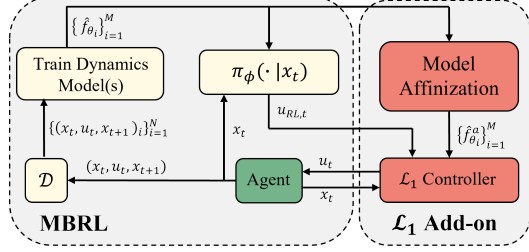

Figure 1: $\mathcal{L}_1$-MBRL Framework. The policy box $\pi_\phi(\cdot|x_t)$ includes policy update and control input sampling for each time step. Although this figure illustrates an on-policy MBRL setting with a parameterized $\pi_\phi$ to provide a simple visualization, the framework is not limited to such class and can also be applied to off-policy algorithms or without a parameterized policy.

tion used to train the baseline MBRL algorithm among other possibilities. This is only for convenience in explaining the $\mathcal{L}_1$ augmentation in Section 3.2, and appropriate adjustments can be readily made for the augmentation upon the choice of different loss functions.

Besides the loss function, methods like random initialization of parameters, varying model architectures, and mini-batch shuffling are widely used to reduce the correlation among the outputs of different models in the ensemble. Further, various standard techniques including early stopping, input/output normalization, and weight normalization can be used to avoid overfitting.

Once the model is learned, control input can be computed by any of the following options: 1) using the learned dynamics as a simulator to generate fictitious samples (Kurutach et al., 2018; Clavera et al., 2018), 2) leveraging the derivative of the model for policy search (Levine & Koltun, 2013; Heess et al., 2015), or 3) applying the Model Predictive Controller (MPC) (Nagabandi et al., 2018; Chua et al., 2018). We highlight here that our proposed method is agnostic to the use of particular techniques or the choice of the policy optimizer.

### 3.2 $\mathcal{L}_1$ AUGMENTATION

Let the true dynamics in discrete time be given by

$$\Delta x_{t+1} = f(x_t, u_t) + w(t, x_t, u_t), \quad \Delta x_{t+1} \triangleq x_{t+1} - x_t, \tag{8}$$

where the transition function $f$ and the system uncertainty $w$ are unknown. Let $\hat{f}_\theta := \frac{1}{M}\sum_{i=1}^M f_{\theta_i}$ be the mean of the ensemble model trained using the loss function in Equation (7). Then, we express

$$\Delta\bar{x}_{t+1} = \hat{f}_\theta(x_t, u_t), \quad \Delta\bar{x}_{t+1} \triangleq \bar{x}_{t+1} - x_t, \tag{9}$$

where $\bar{x}_{t+1}$ indicates the estimate of the next state evaluated with $\hat{f}_\theta(x_t, u_t)$. In MBRL, such transition functions are typically modeled using fully nonlinear function approximators like NNs. However, as discussed in Sec. 2.2, it is necessary to represent the nominal model in the control-affine form to apply $\mathcal{L}_1$ adaptive control. A common approach to obtain a control-affine model involves restricting the model structure to the control-affine class (Khojasteh et al., 2020; Taylor et al., 2019; Choi et al., 2020). For NN models, this process involves training two NNs $g_\theta$ and $h_\theta$, such that Equation (9) becomes $\Delta\bar{x}_{t+1} = g_\theta(x_t) + h_\theta(x_t)u_t$.

While control-affine models are used for their tractability and direct connection to control-theoretic methods, they are inherently limited in their representational power compared to fully nonlinear models, and hence, their use in an MBRL algorithm can result in reduced performance.

To study the level of compromise on the performance, we compare fully nonlinear models with control-affine models in the Halfcheetah environment for METRPO (Kurutach et al., 2018), where each size of the implicit layers of the control-affine model $g_\theta$ and $h_\theta$ are chosen to match that of the fully nonlinear $\hat{f}_\theta$. The degraded performance of the control-affine model depicted in Fig. 2 can be primarily attributed to intricate nonlinearities in the environment.

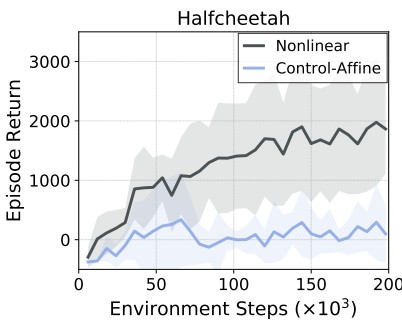

Figure 2: Comparison of performance between fully nonlinear and control-affine model on the Halfcheetah environment using METRPO. The control-affine model failed to learn the Halfcheetah dynamics.

Although using the above naive control-affine model can be convenient, it must trade in the capabilities of the underlying MBRL algorithm. To avoid such limitations, we adopt an alternative approach inspired by the Guided Policy Search (Levine & Koltun, 2013). Specifically, we apply a control-affine transformation to the fully nonlinear dynamics multiple times according to the predefined *switching law*. Specifically, we apply the first-order Taylor series approximation around the operating input $\bar{u}$:

$$\hat{f}_\theta(x_t, u_t) \approx \hat{f}_\theta(x_t, \bar{u}) + \left(\left[\nabla_u \hat{f}_\theta(x_t, u)\right]_{u=\bar{u}}\right)^\top (u_t - \bar{u})$$

$$= \underbrace{\hat{f}_\theta(x_t, \bar{u}) - \left(\left[\nabla_u \hat{f}_\theta(x_t, u)\right]_{u=\bar{u}}\right)^\top \bar{u}}_{\triangleq\, g_\theta(x_t)} + \underbrace{\left(\left[\nabla_u \hat{f}_\theta(x_t, u)\right]_{u=\bar{u}}\right)^\top u_t}_{\triangleq\, h_\theta(x_t)} \triangleq \hat{f}_\theta^a(x_t, u_t; \bar{u}). \quad (10)$$

Here, the superscript $a$ indicates the *affine* approximation of $\hat{f}_\theta$. The term *affinization* in this paper is distinguished from *linearization*, which linearizes the function with respect to both $x_t$ and $u_t$ such that $\bar{x}_{t+1} \simeq Ax_t + Bu_t$ for some constant matrix $A$ and $B$. Since it is common to have more states than control inputs in most controlled systems, the affinized model is a significantly more accurate approximation of the nominal dynamics compared to the linearized model.

Indeed, the control-affine model $\hat{f}_\theta^a$ is only a good approximation of $\hat{f}_\theta$ around $\bar{u}$. When the control input deviates considerably from $\bar{u}$, the quality of the approximation deteriorates. To handle this, we produce the next approximation model when the following *switching* condition holds:

$$\|\hat{f}_\theta^a(x_t, u_t; \bar{u}) - \hat{f}_\theta(x_t, u_t)\| \geq \epsilon_a. \quad (11)$$

Here, $\|\cdot\|$ indicates the vector norm, and $\epsilon_a$ is the model tolerance hyperparameter chosen by the user. Note that as $\epsilon_a \to 0$, we make an affine approximation at every point in the input space and we retrieve the original non-affine function $\hat{f}_\theta$.

**Remark 1.** *Although a more intuitive choice for the switching condition would be $\|u_t - \bar{u}\| > \epsilon_a$, we adopt an implicit switching condition (Equation (11)) to explicitly control over the acceptable level of prediction error between $\hat{f}_\theta^a$ and $\hat{f}_\theta$ by specifying the threshold $\epsilon_a$. This approach prevents significant deviation in the performance of the underlying MBRL algorithm, and its utilization is instrumental in establishing the theoretical guarantees of the uncertainty estimation (See Section 3.3).*

Given a locally valid control-affine model $\hat{f}_\theta^a$, we can proceed with the design of the $\mathcal{L}_1$ input by utilizing the discrete-time version of the controller presented in Sec. 2.2. In particular, the state-

predictor, the adaptation law, and the control law are given by

$$\hat{x}_{t+1} = \hat{x}_t + \Delta \hat{x}_t \triangleq \hat{x}_t + \hat{f}_\theta^a(x_t, u_t) + \hat{\sigma}_t + (A_s \tilde{x}_t)\Delta t, \quad \hat{x}_0 = x_0, \; \tilde{x}_t = \hat{x}_t - x_t, \quad (12a)$$

$$\hat{\sigma}_t = -\Phi^{-1}\mu_t, \quad (12b)$$

$$q_t = q_{t-1} + (-Kq_{t-1} + K\hat{\sigma}_{m,t-1})\Delta t, \quad q_0 = 0, \; u_{a,t} = -q_t, \quad (12c)$$

where $K \succ 0$ is an $m \times m$ symmetric matrix that characterizes the first order low pass filter $C(s) = K(s\mathbb{I}_m + K)^{-1}$, discretized in time. Note that Equation (3)-Equation (6) can be considered as zero-order-hold continuous-time signals of discrete signals produced by Equation (12). As such, $\hat{\sigma}_t$ and $\mu_t$ are defined analogously to their counterparts in the continuous-time definitions. In our setting, where prior information about the desired input signal frequency is unavailable, an unbiased choice is to set $K = \omega\mathbb{I}_m$, where $\omega$ is the chosen cutoff frequency. The sampling time $T_s$ is set to $\Delta t$, which corresponds to the time interval at which the baseline MBRL algorithm operates. The algorithm for the $\mathcal{L}_1$ adaptive control is presented in Algorithm 1.

As the underlying MBRL algorithm updates its model $\hat{f}_\theta$, the corresponding control-affine model $\hat{f}_\theta^a$ and $\mathcal{L}_1$ control input $u_a$ are updated sequentially (Algorithm 1). By incorporating the $\mathcal{L}_1$ control augmentation into the MBRL algorithm, we obtain the $\mathcal{L}_1$-MBRL algorithm, as outlined in Algorithm 2. Note that in this algorithm we are adding $u_{RL,t}$ instead of $u_t$ to the dataset. Intuitively, this is to learn the nominal dynamics that remains after uncertainties get compensated by $u_{a,t}$. Similar approach has been employed previously in (Wang & Ba, 2020, Appendix A.1).

Our $\mathcal{L}_1$-MBRL framework makes a control-affine approximation of the learned dynamics, which is itself an approximation of the ground truth dynamics. Such layers of approximations may amplify errors, which may degrade the effect of the $\mathcal{L}_1$ augmentation. In this section, we prove that the $\mathcal{L}_1$ prediction error is bounded, and subsequently, the $\mathcal{L}_1$ controller effectively compensates for uncertainties. To this end, we conduct a continuous-time analysis of the system that is controlled via $\mathcal{L}_1$-MBRL framework which operates in sampled-time (Åström & Wittenmark, 2013). It is important to note that our adaptation law (Equation (5)) holds the estimation for each time interval (zero-order-hold), converting discrete estimates obtained from the MBRL algorithm into a continuous-time signal. Such choice of the adaptation law ensures that the $\mathcal{L}_1$ augmentation is compatible with the discrete MBRL setup, providing the basis for the following analysis.

---

**Algorithm 1:** $\mathcal{L}_1$ ADAPTIVE CONTROL

**Data:** Initialize $\{\hat{x}_t, \; \hat{\sigma}_t\} \leftarrow 0$
Set $\omega$ for $K$ in Equation (12c)
**Function** Control ($u_{RL,t}, x_t, \hat{f}_\theta^a$)**:**
    Prediction error update: $\tilde{x}_t \longleftarrow \hat{x}_t - x_t$
    Uncertainty estimate $\hat{\sigma}_t$
     update: Equation (12b)
    Compute $u_{a,t}$ (Equation (12c))
    $u_t \leftarrow u_{RL,t} + u_{a,t}$
    Update $\hat{x}_{t+1}$ (Equation (12a))
    **return** $u_t$
**End Function**

---

**Algorithm 2:** $\mathcal{L}_1$-MBRL ALGORITHM

Set $\mathcal{D} \leftarrow \emptyset$, $\{\hat{x}_t, \; \hat{\sigma}_t\} \leftarrow 0$
Initialize $\epsilon, \pi_\phi, \hat{f}_\theta, \omega, x_0$
**repeat**
    **for** *Episodes* $N_e \in \mathbb{N}$ **do**
        $\bar{u} \leftarrow$ None
        **for** *Horizon* $H \in \mathbb{N}$ **do**
            $u_{RL,t} \sim \pi_\phi(\cdot|x_t)$
            **if** $\bar{u}$ *is* None *or Equation (11)* **then**
                $\bar{u} \leftarrow u_{RL,t}$
                compute $\hat{f}_\theta^a$ via Equation (10)
            $u_t \leftarrow$ Control in Algo. 1
            Execute $u_t$ and
                $\mathcal{D} \leftarrow (x_t, u_{RL,t}, x_{t+1})$
    Update model(s) $\hat{f}_\theta$ using $\mathcal{D}$
    Update policy $\pi_\phi$
**until** *the average return converges*

---

### 3.3 THEORETICAL ANALYSIS

Consider the nonlinear (unknown) continuous-time counterpart of Equation (8)

$$\dot{x}(t) = F(x(t), u(t)) + W(t, x(t), u(t))^1, \quad (13)$$

where $F : \mathcal{X} \times \mathcal{U} \to \mathbb{R}^n$ is a fully nonlinear function defining the vector field. Note that unlike the system in Equation (1), we do not make any assumptions on $F(x(t), u(t))$ being control-affine. Furthermore, recall from Sec. 2.1 that the sets $\mathcal{X} \subset \mathbb{R}^n$ and $\mathcal{U} \subset \mathbb{R}^m$, over which the MBRL

---

[1]The continuous-time functions correspond to the Euler-integral of its discrete counterparts.

experiments take place, are compact. Additionally, $W(t, x(t), u(t)) \in \mathbb{R}^n$ represents the disturbance perturbing the system. As before, we denote by $\hat{F}_\theta(x(t), u(t))$ the approximation of $F(x(t), u(t))$, and its affine approximate as

$$\hat{F}_\theta^a(x(t), u(t)) = G_\theta(x(t)) + H_\theta(x(t))u(t). \tag{14}$$

Subsequently, we define the residual error $l(t, x(t), u(t))$ and an affinization error $a(x(t), u(t))$ as

$$l(t, x(t), u(t)) \triangleq F(x(t), u(t)) + W(t, x(t), u(t)) - \hat{F}_\theta(x(t), u(t))$$
$$a(x(t), u(t)) \triangleq \hat{F}_\theta(x(t), u(t)) - \hat{F}_\theta^a(x(t), u(t)).$$

Note that $\|a(x(t), u(t))\| \leq \epsilon_a$ in the $\mathcal{L}_1$-MBRL framework by Equation (11).

We pose the following assumptions.

**Assumption 1.**

1. *The functions $F(x(t), u(t))$ and $W(t, x(t), u(t))$ are Lipschitz continuous over $\mathcal{X} \times \mathcal{U}$ and $[0, t_{\max}) \times \mathcal{X} \times \mathcal{U}$, respectively, for $0 < t_{\max} \leq H$, where $H$ is a known finite time horizon of the episode. The learned model $\hat{F}_\theta(x(t), u(t))$ is Lipschitz continuous in $\mathcal{X}$, and continuously differentiable[2] ($\mathcal{C}^1$) in $\mathcal{U}$.*

2. *The learned model is uniformly bounded over $(t, x, u) \in [0, t_{\max}) \times \mathcal{X} \times \mathcal{U}$:*

$$\|F(x(t), u(t)) + W(t, x(t), u(t)) - \hat{F}_\theta(x(t), u(t))\| \leq \epsilon_l, \tag{15}$$

   *where the bound $\epsilon_l$ is assumed to be known.*

See Appendix B for remarks on this assumption.

Next, we set

$$u(t) = u^\star(t) + u_a(t), \tag{16}$$

where $u^\star(t)$ is the continuous-time signal converted from the discrete control input produced by the underlying MBRL, and $u_a(t)$ is the $\mathcal{L}_1$ control input. As described in Section 2.2, the $\mathcal{L}_1$ controller estimates the uncertainty by following the piecewise constant adaptive law (Equation (5)). Now, we aim to evaluate the estimation error $e(t, x(t), u(t))$:

$$e(t, x(t), u(t)) \triangleq l(t, x(t), u(t)) + a(x(t), u(t)) - \hat{\sigma}(t)$$
$$= F(x(t), u(t)) + W(t, x(t), u(t)) - G_\theta(x(t)) - H_\theta(x(t))u(t) - \hat{\sigma}(t). \tag{17}$$

Our interest in evaluating the estimation error is articulated in Remark 2, Appendix C, where we also provide proof of the following theorem. We note here that the sets $\mathcal{X}$ and $\mathcal{U}$ in the following result are compact due to the inherent nature of the MBRL algorithm, as described in Sec. 2.1.

**Theorem 1.** *Consider the system described by Equation (13), and its learned control-affine representation in Equation (14). Additionally, assume that the system is operating under the augmented feedback control presented in Equation (16). Let $A_s = \texttt{diag}\{\lambda_1, \ldots, \lambda_n\}$ be the Hurwitz matrix that is used in the definition of the state predictor (Equation (3)). If Assumption 1 holds, then the estimation error defined in Equation (17) satisfies $\|e(t, x(t), u(t)\| \leq \epsilon_l + \epsilon_a, \forall t \in [0, T_s)$ and*

$$\|e(t, x(t), u(t))\| = 2\epsilon_a + \mathcal{O}(T_s) \quad \forall t \in [T_s, t_{\max}),$$

*where $0 < T_s < t_{\max} \leq H < \infty$, and $H$ is the known bounded horizon (see Sec. 2.1).*

## 4   SIMULATION RESULTS

In this section, we demonstrate the efficacy of our proposed $\mathcal{L}_1$-MBRL framework using the METRPO (Kurutach et al., 2018) and MBMF (Nagabandi et al., 2018) as the baseline MBRL method [3], and we defer the $\mathcal{L}_1$-MBMF derivations and details to Appendix D. We briefly note here that we observed a similar performance improvement with $\mathcal{L}_1$ augmentation as for $\mathcal{L}_1$-METRPO.

---

[2]More precisely, $\mathcal{C}^1$ everywhere except finite sets of measure zero.

[3]METRPO and MBMF conform to the standard MBRL framework but employ different strategies for control optimization (refer to Section 2.1). Selecting these baselines demonstrates that our framework is agnostic to various control optimization methods, illustrating its functionality as a versatile add-on module.

In our first experimental study, we evaluate the proposed $\mathcal{L}_1$-MBRL framework on five different OpenAI Gym environments (Brockman et al., 2016) with varying levels of state and action complexity. For each environment, we report the mean and standard deviation of the average reward per episode across multiple random seeds. Additionally, we incorporate noise into the observation ($\sigma_o = 0.1$) or action ($\sigma_a = 0.1$) by sampling from a uniform distribution (Wang et al., 2019). This enables us to evaluate the impact of noise on MBRL performance. The results are summarized in Table 1. Further details of the experiment setup are provided in the Appendix D.

Table 1: Performance comparison between METRPO and $\mathcal{L}_1$-METRPO (Ours). The average performance and standard deviation over multiple seeds are evaluated for a window size of 3000 timesteps at the end of the training for multiple seeds. Higher performance cases are marked in bold and green.

| Env. | Noise-free | | $\sigma_{\mathbf{a}} = \mathbf{0.1}$ | | $\sigma_{\mathbf{o}} = \mathbf{0.1}$ | |
|---|---|---|---|---|---|---|
| | METRPO | $\mathcal{L}_1$-METRPO | METRPO | $\mathcal{L}_1$-METRPO | METRPO | $\mathcal{L}_1$-METRPO |
| Inv. P. | $-51.3 \pm 67.8$ | $\mathbf{-0.0 \pm 0.0}$ | $-105.2 \pm 81.6$ | $\mathbf{-0.0 \pm 0.0}$ | $-74.22 \pm 74.5$ | $\mathbf{-21.3 \pm 20.7}$ |
| Swimmer | $309.5 \pm 49.3$ | $\mathbf{313.8 \pm 18.7}$ | $258.7 \pm 113.7$ | $\mathbf{322.7 \pm 5.3}$ | $30.7 \pm 56.1$ | $\mathbf{79.2 \pm 85.0}$ |
| Hopper | $1140.1 \pm 552.4$ | $\mathbf{1491.4 \pm 623.8}$ | $609.0 \pm 793.5$ | $\mathbf{868.7 \pm 735.8}$ | $-1391.2 \pm 266.5$ | $\mathbf{-486.6 \pm 459.9}$ |
| Walker | $\mathbf{-6.6 \pm 0.3}$ | $-6.9 \pm 0.5$ | $-9.8 \pm 2.2$ | $\mathbf{-5.9 \pm 0.3}$ | $-30.3 \pm 28.2$ | $\mathbf{-6.3 \pm 0.3}$ |
| Halfcheetah | $2367.3 \pm 1274.5$ | $\mathbf{2588.6 \pm 955.1}$ | $1920.3 \pm 932.4$ | $\mathbf{2515.9 \pm 1216.4}$ | $1419.0 \pm 517.2$ | $\mathbf{1906.3 \pm 972.7}$ |

The experimental results demonstrate that our proposed $\mathcal{L}_1$-MBRL framework outperforms the baseline METRPO in almost every scenario. Notably, the advantages of the $\mathcal{L}_1$ augmentation become more apparent under noisy conditions.

## 4.1 ABLATION STUDY

We conduct an ablation study to compare the specific contributions of $\mathcal{L}_1$ control in the training and testing. During testing, $\mathcal{L}_1$ control explicitly rejects system uncertainties and improves performance. On the other hand, during training, $\mathcal{L}_1$ additionally influences the learning by shifting the training dataset distribution. To evaluate the effect of $\mathcal{L}_1$ at each phase, we compare four scenarios: 1) no $\mathcal{L}_1$ during training or testing, 2) $\mathcal{L}_1$ applied only during testing, 3) $\mathcal{L}_1$ used only during training, and 4) $\mathcal{L}_1$ applied during both training and testing. The results are summarized in Fig. 3.

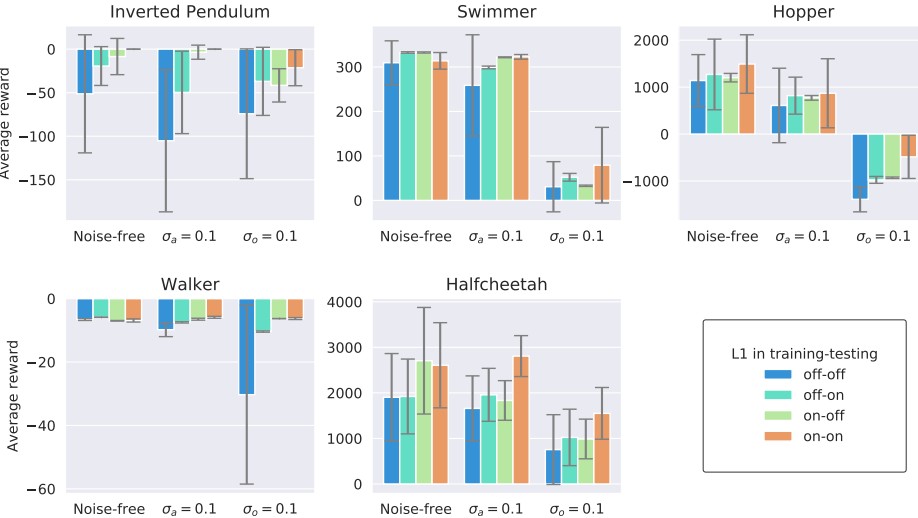

Figure 3: Contribution of $\mathcal{L}_1$ in the training and testing phase. The notation $\mathcal{L}_1$ on (off)-on (off) indicates $\mathcal{L}_1$ is applied (not applied) during training-testing, respectively. The error bar ranges for one standard deviation of the performance. On-on and off-off correspond to our main result in Table 1. As expected, the on-on case achieved the highest performance in most scenarios.

The results depicted in Fig. 3 demonstrate that the influence of $\mathcal{L}_1$ control during training and testing varies across different environments and noise types. However, as anticipated, the highest performance is achieved when $\mathcal{L}_1$ is applied in both the training and testing phases.

In order to evaluate the effectiveness of the $\mathcal{L}_1$ -MBRL framework in addressing the sim-to-real gap, we conducted a secondary ablation study. First, we trained the model without $\mathcal{L}_1$ in a noise-free environment and subsequently tested the model with and without $\mathcal{L}_1$ under a noisy environment. The results are demonstrated in Table 2. This result indicates that our $\mathcal{L}_1$-MBRL framework effectively addresses the sim-to-real gap, and this demonstrates the potential for directly extending our framework to the offline MBRL setting, presenting promising opportunities for future research.

Table 2: Addressing the sim-to-real gap with $\mathcal{L}_1$ augmentation: The original METRPO was initially trained on an environment without uncertainty. Subsequently, the policy was deployed in a noisy environment that emulates real-world conditions, with and without $\mathcal{L}_1$ augmentation.

| Env. | $\sigma_\mathbf{a} = 0.1$ | | $\sigma_\mathbf{o} = 0.1$ | | $\sigma_\mathbf{a} = 0.1$ & $\sigma_\mathbf{o} = 0.1$ | |
| --- | --- | --- | --- | --- | --- | --- |
| | METRPO | $\mathcal{L}_1$-METRPO | METRPO | $\mathcal{L}_1$-METRPO | METRPO | $\mathcal{L}_1$-METRPO |
| Inv. P. | $30.2 \pm 45.1$ | $\mathbf{-0.0 \pm 0.0}$ | $-74.1 \pm 53.1$ | $\mathbf{-3.1 \pm 2.0}$ | $-107.0 \pm 72.4$ | $\mathbf{-6.1 \pm 4.6}$ |
| Swimmer | $250.8 \pm 130.2$ | $\mathbf{330.5 \pm 5.7}$ | $\mathbf{337.8 \pm 2.9}$ | $331.2 \pm 8.34$ | $248.2 \pm 133.6$ | $\mathbf{327.3 \pm 6.8}$ |
| Hopper | $198.9 \pm 617.8$ | $\mathbf{623.4 \pm 405.6}$ | $-84.5 \pm 1035.8$ | $\mathbf{157.1 \pm 379.7}$ | $87.5 \pm 510.2$ | $\mathbf{309.8 \pm 477.8}$ |
| Walker | $\mathbf{-6.0 \pm 0.8}$ | $-6.3 \pm 0.7$ | $-6.4 \pm 0.4$ | $\mathbf{-6.08 \pm 0.6}$ | $-6.3 \pm 0.4$ | $\mathbf{-5.2 \pm 1.5}$ |
| Halfcheetah | $1845.8 \pm 600.9$ | $\mathbf{1965.3 \pm 839.5}$ | $1265.0 \pm 440.8$ | $\mathbf{1861.6 \pm 605.5}$ | $1355.0 \pm 335.6$ | $\mathbf{1643.6 \pm 712.5}$ |

## 5 LIMITATIONS

**(Performance of the base MBRL)** Our $\mathcal{L}_1$-MBRL scheme rejects uncertainty estimates derived from the *learned* nominal dynamics. As a result, the performance of $\mathcal{L}_1$-MBRL is inherently tied to the baseline MBRL algorithm, and $\mathcal{L}_1$ augmentation cannot independently guarantee good performance. This can be related to the role of $\epsilon_l$ in Equation (15). Empirical evidence in Fig. 3 illustrates this point that, despite meaningful improvements, the performance of scenarios augmented with $\mathcal{L}_1$ is closely correlated to that of METRPO without $\mathcal{L}_1$ augmentation.

**(Trade-off in choosing $\epsilon_a$)** In Sec. 3.2, we mentioned that as $\epsilon_a$ approaches zero, the baseline MBRL is recovered. This implies that for small values of $\epsilon_a$, the robustness properties exhibited by the $\mathcal{L}_1$ control are compromised. Conversely, if $\epsilon_a$ is increased excessively, it permits significant deviations between the control-affine and nonlinear models, potentially allowing for larger errors in the state predictor (see Section 3.3). Our heuristic observation from the experiments is to select an $\epsilon_a$ that results in approximately 0-100 affinization switches per 1000 time steps for systems with low complexity ($n < 5$) and 200-500 switches for more complex systems.

## 6 CONCLUSION

In this paper, we proposed an $\mathcal{L}_1$ -MBRL control theoretic add-on scheme to robustify MBRL algorithms against model and environment uncertainties. We affinize the trained nonlinear model according to a switching rule along the input trajectory, enabling the use of $\mathcal{L}_1$ adaptive control. Without perturbing the underlying MBRL algorithm, we were able to improve the overall performance in almost all scenarios with and without aleatoric uncertainties.

The results open up interesting research directions where we wish to test the applicability of $\mathcal{L}_1$-MBRL on offline MBRL algorithms to address the sim-to-real gap (Kidambi et al., 2020; Yu et al., 2020). Moreover, its consolidation with a distributionally robust optimization problem to address the distribution shift is of interest. Finally, we will also research the $\mathcal{L}_1$-MBRL design for MBRL algorithms with probabilistic models (Chua et al., 2018; Wang & Ba, 2020) to explore a method to utilize the covariance information in addition to mean dynamics.

ACKNOWLEDGMENTS

This work is financially supported by National Aeronautics and Space Administration (NASA) ULI (80NSSC22M0070), NASA USRC (NNH21ZEA001N-USRC), Air Force Office of Scientific Research (FA9550-21-1-0411), National Science Foundation (NSF) AoF Robust Intelligence (2133656), NSF CMMI (2135925), and NSF SLES (2331878).

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

## APPENDIX

## A    EXTENDED DESCRIPTION OF $\mathcal{L}_1$ ADAPTIVE CONTROL

In this section, we provide a detailed explanation of $\mathcal{L}_1$ adaptive control. Consider the following system dynamics:

$$\dot{x}(t) = g(x(t)) + h(x(t))u(t) + d(t, x(t), u(t)), \quad x(0) = x_0, \tag{18}$$

where $x(t) \in \mathbb{R}^n$ is the system state vector, $u(t) \in \mathbb{R}^m$ is the control signal, $g : \mathbb{R}^n \to \mathbb{R}^n$ and $h : \mathbb{R}^n \to \mathbb{R}^{n \times m}$ are known functions that define the desired dynamics, both of which are locally-Lipschitz continuous functions. Furthermore, $d(t, x(t), u(t)) \in \mathbb{R}^n$ represents the unknown nonlinearities and is continuous in its arguments. We now decompose $d(t, x(t), u(t))$ with respect to the range and kernel of $h(x(t))$ to obtain

$$\dot{x}(t) = g(x(t)) + h(x(t))(u(t) + \sigma_m(t, x(t), u(t))) + h^\perp(x(t))\sigma_{um}(t, x(t), u(t)), \quad x(0) = x_0, \tag{19}$$

where $h(x(t))\sigma_m(t, x(t), u(t)) + h^\perp(x(t))\sigma_{um}(t, x(t), u(t)) = d(t, x(t), u(t))$. Moreover, $h^\perp(x(t)) \in \mathbb{R}^{n \times (n-m)}$ denotes a matrix whose columns are perpendicular to $h(x(t)) \in \mathbb{R}^{n \times m}$, such that $h(x(t))^\top h^\perp(x(t)) = 0$ for any $x(t) \in \mathbb{R}^n$. The existence of $h^\perp(x(t))$ is guaranteed if it is a full-rank matrix. The terms $\sigma_m(t, x(t), u(t))$ and $\sigma_{um}(t, x(t), u(t))$ are commonly referred to as *matched* and *unmatched* uncertainties, respectively.

Consider the *nominal* system in the absence of uncertainties

$$\dot{x}^\star(t) = g(x^\star(t)) + h(x^\star(t))u^\star(t), \quad x^\star(0) = x_0,$$

where $u^\star(t)$ is the baseline input designed so that the desired performance and safety requirements are satisfied. If we pass the baseline input to the true system in Equation (18), the actual state trajectory $x(t)$ can diverge from the desired state trajectory $x^\star(t)$ in an unquantifiable manner due to the presence of uncertainties $d(t, x(t), u(t))$. To avoid this behavior, we employ $\mathcal{L}_1$ adaptive control, which aims to compute an input $u_a(t)$ such that, when combined with the nominal input $u^\star(t)$, forms the *augmented input*

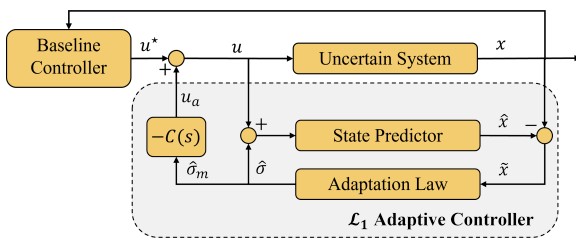

Figure 4: The architecture of $\mathcal{L}_1$ adaptive controller.

$$u(t) = u^\star(t) + u_a(t). \tag{20}$$

The objective of this approach is to ensure that the true state $x(t)$ remains uniformly and quantifiably bounded around the nominal trajectory $x^\star(t)$ under certain conditions.

Next, we explain how the $\mathcal{L}_1$ adaptive controller achieves this goal. The $\mathcal{L}_1$ adaptive control has three components: the state predictor, the adaptation law, and a low-pass filter. The *state predictor* for Equation (18) is given by

$$\dot{\hat{x}}(t) = g(x(t)) + h(x(t))u(t) + \hat{\sigma}(t) + A_s\tilde{x}(t), \tag{21}$$

where $\hat{\sigma}(t) \triangleq h(x(t))\hat{\sigma}_m(t) + h^\perp(x(t))\hat{\sigma}_{um}(t)$. Moreover, $\hat{\sigma}_m(t)$ and $\hat{\sigma}_{um}(t)$ are the estimates of the matched and unmatched uncertainties $\sigma_m(t, x(t))$ and $\sigma_{um}(t, x(t))$, respectively. The initial conditions are given by $\hat{x}(0) = x_0$, $\hat{\sigma}_m(0) = 0$, and $\hat{\sigma}_{um}(0) = 0$. Here, $\hat{x}(t) \in \mathbb{R}^n$ is the state of the predictor, $u(t) \in \mathbb{R}^m$ is the augmented control input in Equation (20), $\tilde{x}(t) = \hat{x}(t) - x(t)$ denotes the *prediction error*, and $A_s \in \mathbb{S}^n$ is a chosen Hurwitz matrix. The state predictor produces the state estimate $\hat{x}(t)$ induced by the adaptive estimates $\hat{\sigma}_m(t)$ and $\hat{\sigma}_{um}(t)$.

Following the true dynamics in Equation (18) and the state predictor in Equation (21), the dynamics of the prediction error $\tilde{x}(t) = \hat{x}(t) - x(t)$ is given by

$$\dot{\tilde{x}}(t) = A_s\tilde{x}(t) + [\hat{\sigma}(t) - d(t, x(t), u(t))], \quad \tilde{x}(0) = 0. \tag{22}$$

Here, we refer to $\hat{\sigma}(t) - d(t, x(t), u(t))$ as the *uncertainty estimation error*. Moreover, since $A_s$ is Hurwitz, the system Equation (22) governing the evolution of $\tilde{x}(t)$ is exponentially stable in the absence of the exogenous input $\hat{\sigma}(t) - d(t, x(t), u(t))$. Therefore, $\tilde{x}(t)$ serves as a *learning signal* for the adaptive law, which we describe next.

We employ a *piecewise constant estimation* scheme (Hovakimyan & Cao, 2010) based on the solution of Equation (22), which can be expressed using the following equation:

$$\tilde{x}(t) = \exp{(A_s t)}\tilde{x}(0) + \int_0^t \exp{(A_s(t - \tau))}(\hat{\sigma}(\tau) - d(\tau, x(\tau), u(\tau))d\tau. \tag{23}$$

For a given sampling time $T_s > 0$, we use a piecewise constant estimate defined as

$$\hat{\sigma}(t) = \hat{\sigma}(iT_s), \quad t \in [iT_s, (i+1)T_s), \ i \in \mathbb{N} \cup \{0\}.$$

When the system is initialized ($i = 0$), we set $\tilde{x}(t) = 0$ which implies $\hat{\sigma}(t) = 0$ for $t \in [0, T_s)$. Now consider a time index $i \in \mathbb{N}$, and the time interval $[iT_s, (i+1)T_s)$. Over this time interval, the solution of Equation (23), obtained by applying the piecewise constant representation, can be written as

$$\tilde{x}(t) = \exp{(A_s(t - iT_s))}\tilde{x}(iT_s)$$
$$+ \int_{iT_s}^t \exp{(A_s(t - \tau))}(\hat{\sigma}(\tau) - d(\tau, x(\tau), u(\tau))d\tau, \quad t \in [iT_s, (i+1)T_s).$$

Now, note that over the previous interval $t \in [(i-1)T_s, iT_s)$ the system was affected by uncertainty $d(t, x(t), u(t))$, which resulted in $\tilde{x}(iT_s) \neq 0$. At the end of the time interval $[iT_s, (i+1)T_s)$, we obtain

$$\tilde{x}((i+1)T_s) = \exp{(A_s T_s)}\tilde{x}(iT_s) + \int_{iT_s}^{(i+1)T_s} \exp{(A_s((i+1)T_s - \tau))}\hat{\sigma}(iT_s)d\tau + R_{(i+1)T_s}$$
$$= \exp{(A_s T_s)}\tilde{x}(iT_s) + A_s^{-1}\left(\exp{(A_s T_s)} - \mathbb{I}_n\right)\hat{\sigma}(iT_s) + R_{(i+1)T_s}, \tag{24}$$

where

$$R_{(i+1)T_s} \triangleq -\int_{iT_s}^{(i+1)T_s} \exp{(A_s((i+1)T_s - \tau))}d(\tau, x(\tau), u(\tau))d\tau$$

is the residual term that captures the uncertainty entered during $[iT_s, (i+1)T_s)$.

Let

$$\hat{\sigma}(iT_s) = -\Phi^{-1}(T_s)\mu(iT_s), \quad \hat{\sigma}(0) = 0, \tag{25}$$

where $\Phi(T_s) = A_s^{-1}(\exp{(A_s T_s)} - \mathbb{I})$ and $\mu(iT_s) = \exp{(A_s T_s)}\tilde{x}(iT_s)$ for $i = 0, 1, 2, \cdots$. Substituting this into Equation (24) removes the first two terms, leaving us only with the residual term, which will appear as the initial condition of the next time interval. In other words, the adaptation law attempts to remove the effect of the uncertainty introduced in the current time interval by addressing it at the start of the subsequent interval. Interested readers can refer to (Kharisov, 2013, Ch. 2) for further details on the piecewise constant adaptive law.

Note that a small sampling time $T_s$ results in a small prediction error $\|\tilde{x}(iT_s)\|$ for each $i = 1, 2, \cdots$. Therefore, it is desirable to keep $T_s$ small up to the hardware limit. However, setting a small $T_s$ and/or having large eigenvalues of $A_s$ can lead to a high adaptation gain ($\Phi^{-1}$ in Equation (25)). This can result in high-frequency uncertainty estimation, which can reduce the robustness of the controlled system if we directly apply $u_a(t) = -\hat{\sigma}_m(t)$ to cancel the estimated matched uncertainty. Therefore, we use a low-pass filter in the controller to decouple the fast estimation from the control loop, allowing us to employ an arbitrarily fast adaptation while maintaining the desired robustness. To be specifc, the input $u_a(t)$ is given by

$$u_a(s) = -C(s)\mathfrak{L}\left[\hat{\sigma}_m(t)\right],$$

where $\mathfrak{L}[\cdot]$ denotes the Laplace transform, and the $C(s)$ is the low-pass filter. The cutoff frequency of the low-pass filter is chosen to satisfy small-gain stability conditions, examples of which can be found in (Wang & Hovakimyan, 2011; Lakshmanan et al., 2020). We refer the interested reader to (Pravitra et al., 2020; Wu et al., 2022; Cheng et al., 2022) for further reading on the design process of the $\mathcal{L}_1$ adaptive control.

## B    REMARKS ON ASSUMPTION 1

It is evident from Equation (10) that our method relies on the continuous differentiability of $\hat{F}_\theta(x(t), u(t))$ with respect to $u(t)$ to ensure the continuity of $\hat{F}_\theta^a(x(t), u(t))$. Such a requirement is readily satisfied when using $\mathcal{C}^1$ (or higher order continuously differentiable) activation functions for $\hat{F}_\theta(x(t), u(t))$, such as `sigmoid`, `tanh`, or `swish`. For MBRL algorithms that use activation functions that are not $\mathcal{C}^1$, we can skip the switching law (Equation (11)) to avoid making affine approximations at non-differentiable points (e.g., the origin for `ReLU`).

The first part of Assumption 1 on the regularity of $F(x(t), u(t))$ and $W(t, x(t))$ is standard to ensure the well-posedness (uniqueness and existence of solutions) for Equation (13) (Khalil, 2002, Theorem 3.1). Furthermore, as stated above, since $\hat{F}_\theta(x(t), u(t))$ is $\mathcal{C}^1$ in its arguments, its derivative $\hat{F}_\theta(x(t), u(t))$ is continuous and hence, satisfies the local Lipschitz continuity over the compact sets $\mathcal{X}$ and $\mathcal{U}$ trivially.

The assumption in Equation (15) is satisfied owing to the Lipschitz continuity of the function in its respective arguments. If the bound is unknown, it is possible to collect data by interacting with the environment under a specific policy and initial condition, and then compute a probabilistic bound. However, such a bound is applicable only to the chosen data set and may not hold for other choices of samples, which is the well-known issue of distribution shift (Quinonero-Candela et al., 2008). This assumption, which explicitly bounds the unknown component of the dynamics, although cannot be guaranteed in the real environment, is commonly made in assessing theoretical guarantees of error propagation when using learned models, as seen in previous papers (Knuth et al., 2021; Manzano et al., 2020; Koller et al., 2018).

## C    PROOF OF THEOREM 1

*Proof of Theorem 1.* We begin by applying the triangle inequality for the estimation error:

$$
\begin{aligned}
&\|e(t, x(t), u(t))\| \\
&= \|F(x(t), u(t)) + W(t, x(t), u(t)) - G_\theta(x(t)) - H_\theta(x(t))u(t) - \hat{\sigma}(t)\| \\
&= \|\hat{F}_\theta(x, u(t)) + l(t, x(t), u(t)) - \hat{F}_\theta^a(x(t), u(t)) - \hat{\sigma}(t)\| \\
&\leq \|\hat{F}_\theta(x(t), u(t)) - \hat{F}_\theta^a(x(t), u(t))\| + \|l(t, x(t), u(t)) - \hat{\sigma}(t)\| \\
&\leq \epsilon_a + \|l(t, x(t), u(t)) - \hat{\sigma}(t)\|, \quad i \in \{0\} \cup \mathbb{N},
\end{aligned}
\tag{26}
$$

where we used Equation (11).

For the case when $i = 0$, due to the initial conditions $\tilde{x}(0) = 0$, $\hat{\sigma}(0) = 0$, and the assumption in Equation (15), we get that

$$
\|l(t, x(t), u(t)) - \hat{\sigma}(0)\| = \|l(t, x(t), u(t))\| \leq \epsilon_l, \quad \forall t \in [0, T_s).
$$

Substituting this expression into Equation (26) proves the stated result for $t \in [0, T_s)$.

Next, we bound the term $\|l(t, x(t), u(t)) - \hat{\sigma}(iT_s)\|$ in Equation (26) for all $t \in [T_s, t_{\max})$. Consider an $i \in \mathbb{N}$ that corresponds to $t \in [T_s, t_{\max})$, i.e., $i \in \{1, \ldots, \lfloor t_{\max}/T_s \rfloor\} \triangleq \mathcal{I}$. For any such $i$, substituting the adaptation law from Equation (25) into Equation (24) for the interval $[(i-1)T_s, iT_s)$ produces the following expression

$$
\tilde{x}(iT_s) = -\int_{(i-1)T_s}^{iT_s} \exp(A_s(iT_s - \tau))d(\tau, x(\tau), u(\tau))d\tau, \quad \forall i \in \mathcal{I}.
\tag{27}
$$

Replacing $d(\tau, x(\tau), u(\tau)) = [d_1(\tau, x(\tau), u(\tau)) \quad \ldots \quad d_n(\tau, x(\tau), u(\tau))]^\top$ in Equation (27) and, by the definition of $A_s$ in the theorem statement, we obtain

$$
\tilde{x}(iT_s) = -\int_{(i-1)T_s}^{iT_s} \begin{bmatrix} \exp(\lambda_1(iT_s - \tau))d_1(\tau, x(\tau), u(\tau)) \\ \vdots \\ \exp(\lambda_n(iT_s - \tau))d_n(\tau, x(\tau), u(\tau)) \end{bmatrix} d\tau,
$$

for all $i \in \mathcal{I}$. For brevity, we denote $d_j(\tau) = d_j(\tau, x(\tau), u(\tau))$ for $j = 1, 2, \ldots, n$.

Since $d_j(t)$ is continuous due to Assumption 1 and $\exp(A_s(iT_s - \tau))$ is positive semi-definite, we invoke the Mean Value Theorem (Mendelson, 2022, Sec. 24.1) element wise. We conclude that there exist $t_{c_j} \in [(i-1)T_s, iT_s)$ for each $j \in \{1, \ldots, n\}$ such that

$$\tilde{x}(iT_s) = - \begin{bmatrix} \int_{(i-1)T_s}^{iT_s} \exp(\lambda_1(iT_s - \tau))d_1(t_{c_1})d\tau \\ \vdots \\ \int_{(i-1)T_s}^{iT_s} \exp(\lambda_n(iT_s - \tau))d_n(t_{c_n})d\tau \end{bmatrix}$$

$$= \begin{bmatrix} \frac{1}{\lambda_1}(1 - \exp(\lambda_1 T_s))d_1(t_{c_1}) \\ \vdots \\ \frac{1}{\lambda_n}(1 - \exp(\lambda_n T_s))d_n(t_{c_n}) \end{bmatrix}. \tag{28}$$

Substituting Equation (28) into Equation (25) gives

$$\hat{\sigma}(t) = \hat{\sigma}(iT_s) = \begin{bmatrix} \exp(\lambda_1 T_s)d_1(t_{c_1}) \\ \vdots \\ \exp(\lambda_n T_s)d_n(t_{c_n}) \end{bmatrix}, \tag{29}$$

which is the piece-wise constant estimate of the uncertainty for $t \in [iT_s, (i+1)T_s)$.

Next, using the piece-wise constant adaptation law from Equation (29), we bound $\|l(t, x(t), u(t)) - \hat{\sigma}(t)\|$, for $t \in [iT_s, (i+1)T_s)$. Since $A_s$ is Hurwitz and diagonal, its diagonal elements satisfy $\lambda_j < 0$, for $j = 1, \ldots, n$. Thus, we have

$$\|l(t, x(t), u(t)) - \hat{\sigma}(t)\| = \left\| l(t, x(t), u(t)) - \begin{bmatrix} \exp(\lambda_1 T_s)d_1(t_{c_1}) \\ \vdots \\ \exp(\lambda_n T_s)d_n(t_{c_n}) \end{bmatrix} \right\|$$

$$= \|l(t, x(t), u(t)) - d(t_c) + (\mathbb{I}_n - \exp(A_s T_s)d(t_c))\|$$

$$\leq \|l(t, x(t), u(t)) - d(t_c)\| + \|\mathbb{I}_n - \exp(A_s T_s)\|\|d(t_c)\|$$

$$\leq \|l(t, x(t), u(t)) - d(t_c)\| + (1 - \exp(\lambda_{\min} T_s))\|d(t_c)\|, \tag{30}$$

where $d(t_c) := [d_1(t_{c_1}) \quad \ldots \quad d_n(t_{c_n})]^\top$, $d(\tau) = l(\tau, x(\tau), u(\tau)) + a(x(\tau), u(\tau))$, and $\lambda_{\min}$ denotes the eigenvalue of $A_s$ that has the minimum absolute value. Using the triangle inequality, we get $\|d(t_c)\| = \|l(t_c, x(t_c), u(t_c) + a(x(t_c), u(t_c))\| \leq \epsilon_l + \epsilon_a$. Thus, Equation (30) can be written as

$$\|l(t, x(t), u(t)) - \hat{\sigma}(t)\| \leq \|l(t, x(t), u(t)) - d(t_c)\| + (1 - \exp(\lambda_{\min} T_s))(\epsilon_l + \epsilon_a). \tag{31}$$

Next, we obtain the upper bound for $\|l(t, x(t), u(t)) - d(t_c)\|$ as

$$\|l(t, x(t), u(t)) - d(t_c)\| = \|l(t, x(t), u(t)) - l(t_c, x(t_c), u(t_c)) - a(x(t_c), u(t_c))\|$$

$$\leq \|l(t, x(t), u(t)) - l(t_c, x(t_c), u(t_c))\| + \|a(x(t_c), u(t_c))\|$$

$$\leq \|l(t, x(t), u(t)) - l(t_c, x(t_c), u(t_c))\| + \epsilon_a, \tag{32}$$

where we used the triangle inequality and Equation (11). Due to the Assumption 1, $l(t, x(t), u(t))$ is Lipschitz over the domain of its arguments. Hence, there exist positive scalars $L_{l,t}, L_{l,x}, L_{l,u}$ such that

$$\|l(t, x(t), u(t)) - l(t_c, x(t_c), u(t_c))\| \leq L_{l,t}|t - t_c| + L_{l,x}\|x(t) - x(t_c)\| + L_{l,u}\|u(t) - u(t_c)\|. \tag{33}$$

Furthermore, due to the compactness of $\mathcal{X}$ and $\mathcal{U}$, there exist $L_{x,t}, L_{u,t}$ such that the following inequalities hold:

$$\|x(t) - x(t_c)\| \leq L_{x,t}|t - t_c|, \quad \|u(t) - u(t_c)\| \leq L_{u,t}|t - t_c|.$$

Substituting these bounds into Equation (33), we get

$$\|l(t, x(t), u(t)) - l(t_c, x(t_c), u(t_c))\| \leq L|t - t_c| \leq LT_s, \tag{34}$$

where $L \triangleq L_{l,t} + L_{l,x}L_{x,t} + L_{l,u}L_{u,t} < \infty$. We proceed by sequentially applying the derived bounds, starting with the substitution of Equation (34) into Equation (32), and then employing the resulting bound in Equation (31). The proof is then concluded by incorporating the final bound into Equation (26) and noting that

$$(1 - \exp(\lambda_{\min} T_s))(\epsilon_l + \epsilon_a) + LT_s \in \mathcal{O}(T_s).$$

$\square$

**Remark 2.** *We provide some insights into the interpretation of this theorem. The theorem serves to quantify the predictive quality of the state predictor in the $\mathcal{L}_1$ add-on scheme in terms of the model approximation errors $\epsilon_l$ and $\epsilon_a$, and the parameters governing the $\mathcal{L}_1$ add-on scheme ($T_s$, $\lambda_{\min}$). As the control input is computed by low-pass filtering the uncertainty estimate, the performance of the $\mathcal{L}_1$ augmentation is inherently tied to its predictive quality. Theorem 1 establishes that the error in state prediction, induced by estimated uncertainty, can be reduced down to $2\epsilon_a$ by reducing the sampling time $T_s$. In other words, we can accurately estimate the learning error $l(t, x(t), u(t))$, with the predictive accuracy being bounded only by the tunable parameter $\epsilon_a$.*

## D   EXTENDED SIMULATION RESULTS

### D.1   EXPERIMENT SETUP

We provide the dimensionality of the selected environments for our simulation analysis in Table 3. For $\mathcal{L}_1$ -METRPO, the number of iterations for each environment was chosen to obtain asymptotic performance, whereas for $\mathcal{L}_1$ -MBMF we fixed the number of iterations to 200K. Such a setup for MBMF is due to the unique structure of MBMF to use MBRL only to serve as the policy initialization, with which MFRL is executed until the performance reaches the asymptotic results. See Appendix D.2.2 for a detailed explanation of MBMF.

Table 3: Dimensions of state and action space of the environments used in the simulations.

| Environment Name | State space dimension ($n$) | Action space dimension ($m$) |
|---|---|---|
| Inverted Pendulum | 4 | 1 |
| Swimmer | 8 | 2 |
| Hopper | 11 | 3 |
| Walker | 17 | 6 |
| Halfcheetah | 17 | 6 |

We adopted the hyperparameters that have been reported to be effective by the baseline MBRL, which in our case are METRPO and MBMF (Wang et al., 2019, Appendix B.4, B.5). Additional hyperparameters introduced by the $\mathcal{L}_1$-MBRL scheme are the affinization threshold $\epsilon$, the cutoff frequency $\omega$, and the Hurwitz matrix $A_s$. Throughout all experiments, we fixed $A_s$ as a negative identity matrix $-\mathbb{I}_n$. For the Inverted Pendulum environment, we set $\epsilon = 1$ and for Halfcheetah $\epsilon = 3$, while for other environments, we chose $\epsilon = 0.3$. Additionally, we selected a cutoff frequency of $\omega = 0.35/T_s$, where $T_s$ represents the sampling time interval of the environment. **It is important to note that the $\mathcal{L}_1$ controller has not been redesigned or retuned through all the experiments.**

### D.2   TECHNICAL REMARKS

If the baseline algorithm employs any data processing techniques such as input/output normalization, as discussed briefly in Section 3.1, our state predictor and controller (Equation (12a),Equation (12c)) must also follow the corresponding process.

### D.2.1   METRPO

METRPO trains an ensemble model, from which fictitious samples are generated. Then, the policy network is updated following the TRPO (Schulman et al., 2015) in the policy improvement step. The input and output of the neural network are normalized during the training step, and consequently, calculation of the Jacobian in $\mathcal{L}_1$ -METRPO must *unnormalize* the result. Specifically, this process is carried out by applying the chain rule, which includes multiplying the normalized Jacobian matrix ($J'$) by the standard deviations of the inputs and outputs given by Equation (35):

$$J = D_{\Delta x} J' D_{x,u}^{-1}, \tag{35}$$

where $D_{\Delta x} = \text{diag}\{\sigma_{\Delta x_1}, \dots, \sigma_{\Delta x_n}\}$ and $D_{x,u} = \text{diag}\{\sigma_{x_1}, \dots, \sigma_{x_n}, \sigma_{u_1}, \dots, \sigma_{u_m}\}$.

This unnormalized Jacobian ($J$) is subsequently utilized to generate the $\mathcal{L}_1$ adaptive control output.

### D.2.2 MBMF

In the MBMF algorithm (Nagabandi et al., 2018), the authors begin by training a Random Shooting (RS) controller. This controller is then distilled into a neural network policy using the supervised framework DAgger (Ross et al., 2011), which minimizes the KL divergence loss between the neural network policy and the RS controller. Then, the policy is fine-tuned using standard model-free algorithms like TRPO (Schulman et al., 2015) or PPO (Schulman et al., 2017). We adopt a similar approach to what was done for METRPO. The Jacobian matrix of the neural network is unnormalized based on Equation (35). The adaptive controller is augmented to the RS controller based on the latest model trained.

### D.3 EXPERIMENT RESULTS

In this section, we first present the results of $\mathcal{L}_1$ -MBMF in comparison to MBMF without $\mathcal{L}_1$ augmentation. The corresponding tabular results are summarized in Table 4. Noticeably, $\mathcal{L}_1$ augmentation improves the MBMF algorithm in *every* case uniformly.

Table 4: Performance comparison between MBMF and $\mathcal{L}_1$-MBMF (Ours). The performance is averaged across multiple random seeds with a window size of 5000 timesteps at the end of the training. Higher performance is written in bold and green.

| Env. | Noise-free | | $\sigma_{\mathbf{a}} = \mathbf{0.1}$ | | $\sigma_{\mathbf{o}} = \mathbf{0.1}$ | |
|---|---|---|---|---|---|---|
| | MB-MF | $\mathcal{L}_1$-MB-MF | MB-MF | $\mathcal{L}_1$-MB-MF | MB-MF | $\mathcal{L}_1$-MB-MF |
| Inv. P. | $-100.5 \pm 4.3$ | $\mathbf{-10.5 \pm 3.7}$ | $-7.4 \pm 1.5$ | $\mathbf{-4.8 \pm 1.9}$ | $-10.2 \pm 2.4$ | $\mathbf{-5.09 \pm 1.6}$ |
| Swimmer | $284.9 \pm 25.1$ | $\mathbf{314.3 \pm 3.3}$ | $304.8 \pm 1.9$ | $\mathbf{314.5 \pm 0.6}$ | $292.8 \pm 1.3$ | $\mathbf{294.3 \pm 4.3}$ |
| Hopper | $-1047.4 \pm 1098.7$ | $\mathbf{350.1 \pm 465.2}$ | $-877.9 \pm 383.4$ | $\mathbf{-285.4 \pm 65.3}$ | $-996.9 \pm 206.0$ | $\mathbf{-171.5 \pm 317.3}$ |
| Walker | $-1743.7 \pm 233.3$ | $\mathbf{-1481.7 \pm 322.9}$ | $-2962.2 \pm 178.6$ | $\mathbf{-2447.4 \pm 329.7}$ | $-3348.8 \pm 210.1$ | $\mathbf{-2261.4 \pm 381}$ |
| Halfcheetah | $126.9 \pm 72.7$ | $\mathbf{304.5 \pm 56.0}$ | $184.0 \pm 148.9$ | $\mathbf{299.8 \pm 61.0}$ | $146.1 \pm 87.8$ | $\mathbf{235.2 \pm 19.2}$ |

Additionally, we provide detailed tabular values corresponding to the results shown in Fig. 3. Table 5 provides a summary of the scenarios where $\mathcal{L}_1$ control is augmented only during either training or testing. The application of $\mathcal{L}_1$ control during the testing phase clearly benefits from the explicit rejection of system uncertainty, leading to performance improvement. On the contrary, when $\mathcal{L}_1$ control is applied during the training phase, it not only mitigates uncertainty along the trajectory but also implicitly affects the training process by inducing a shift in the distribution of the training dataset. This study compares these two types of impact brought about by the $\mathcal{L}_1$ augmentation.

Table 5: Comparison of $\mathcal{L}_1$ augmentation effects during training and testing. $\mathcal{L}_1$ -METRPO (Train) refers to the application of $\mathcal{L}_1$ augmentation solely during training, whereas $\mathcal{L}_1$ -METRPO (Test) indicates training without $\mathcal{L}_1$ augmentation and the application of $\mathcal{L}_1$ only during testing.

| Env. | Noise-free | | $\sigma_{\mathbf{a}} = \mathbf{0.1}$ | | $\sigma_{\mathbf{o}} = \mathbf{0.1}$ | |
|---|---|---|---|---|---|---|
| | $\mathcal{L}_1$-METRPO (Train) | $\mathcal{L}_1$-METRPO(Test) | $\mathcal{L}_1$-METRPO(Train) | $\mathcal{L}_1$-METRPO(Test) | $\mathcal{L}_1$-METRPO(Train) | $\mathcal{L}_1$-METRPO(Test) |
| Inv. P. | $\mathbf{-8.50 \pm 20.75}$ | $-19.36 \pm 22.3$ | $\mathbf{-3.52 \pm 8.08}$ | $-49.72 \pm 47.34$ | $-41.63 \pm 19.11$ | $\mathbf{-37.00 \pm 39.07}$ |
| Swimmer | $332.6 \pm 1.3$ | $332.6 \pm 1.6$ | $\mathbf{321.8 \pm 1.0}$ | $298.9 \pm 3.1$ | $32.9 \pm 1.5$ | $\mathbf{52.0 \pm 8.7}$ |
| Hopper | $1201.2 \pm 90.8$ | $\mathbf{1269.9 \pm 752.9}$ | $771.1 \pm 49.8$ | $\mathbf{818.1 \pm 394.2}$ | $\mathbf{-931.7 \pm 15.4}$ | $-976.8 \pm 73.1$ |
| Walker | $-7.0 \pm 0.1$ | $\mathbf{-5.9 \pm 0.0}$ | $\mathbf{-6.5 \pm 0.3}$ | $-7.5 \pm 0.2$ | $\mathbf{-6.3 \pm 0.0}$ | $-10.4 \pm 0.2$ |
| Halfcheetah | $\mathbf{2706.2 \pm 1170.4}$ | $1921.56 \pm 821.34$ | $1834 \pm 434.87$ | $\mathbf{1957.5 \pm 581.6}$ | $987.90 \pm 435.90$ | $\mathbf{1022.1 \pm 619.8}$ |

Notably, there is no consistent trend regarding whether $\mathcal{L}_1$ control has a greater impact during testing or training phases. The primary conclusion drawn from this ablation study - in conjunction with Fig. 3 - is that $\mathcal{L}_1$ augmentation yields the greatest benefits when applied to both training and testing. One possible explanation for this observation is that such consistent augmentation avoids a shift in the policy distribution, leading to desired performance.

Next, in Fig. 5, we report the learning curves of the main result.

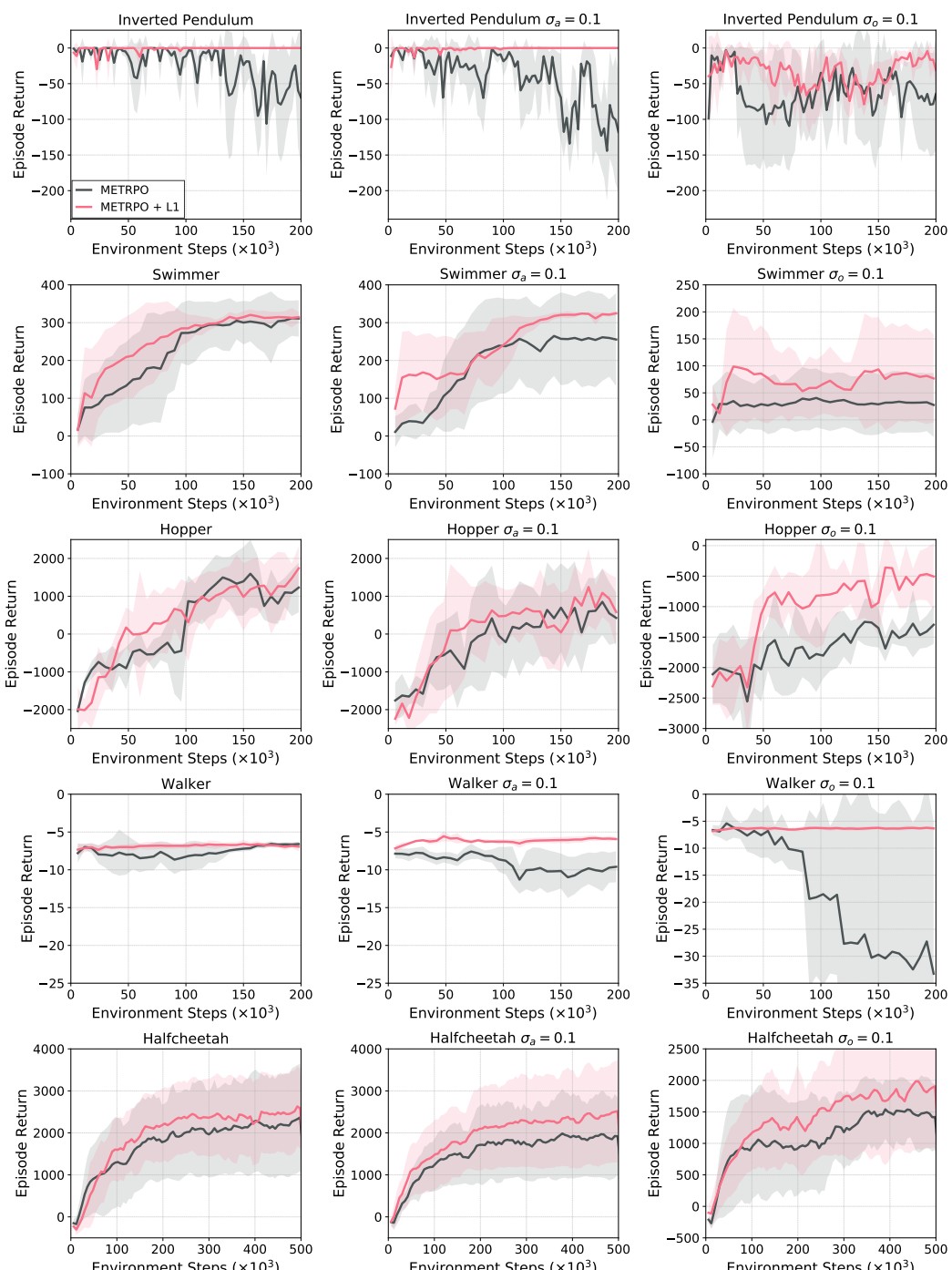

Figure 5: Plots of $\mathcal{L}_1$-METRPO learning curves as a function of episodic steps. The performance is averaged across multiple random seeds such that the solid lines indicate the average return at the corresponding timestep, and the shaded regions indicate one standard deviation.

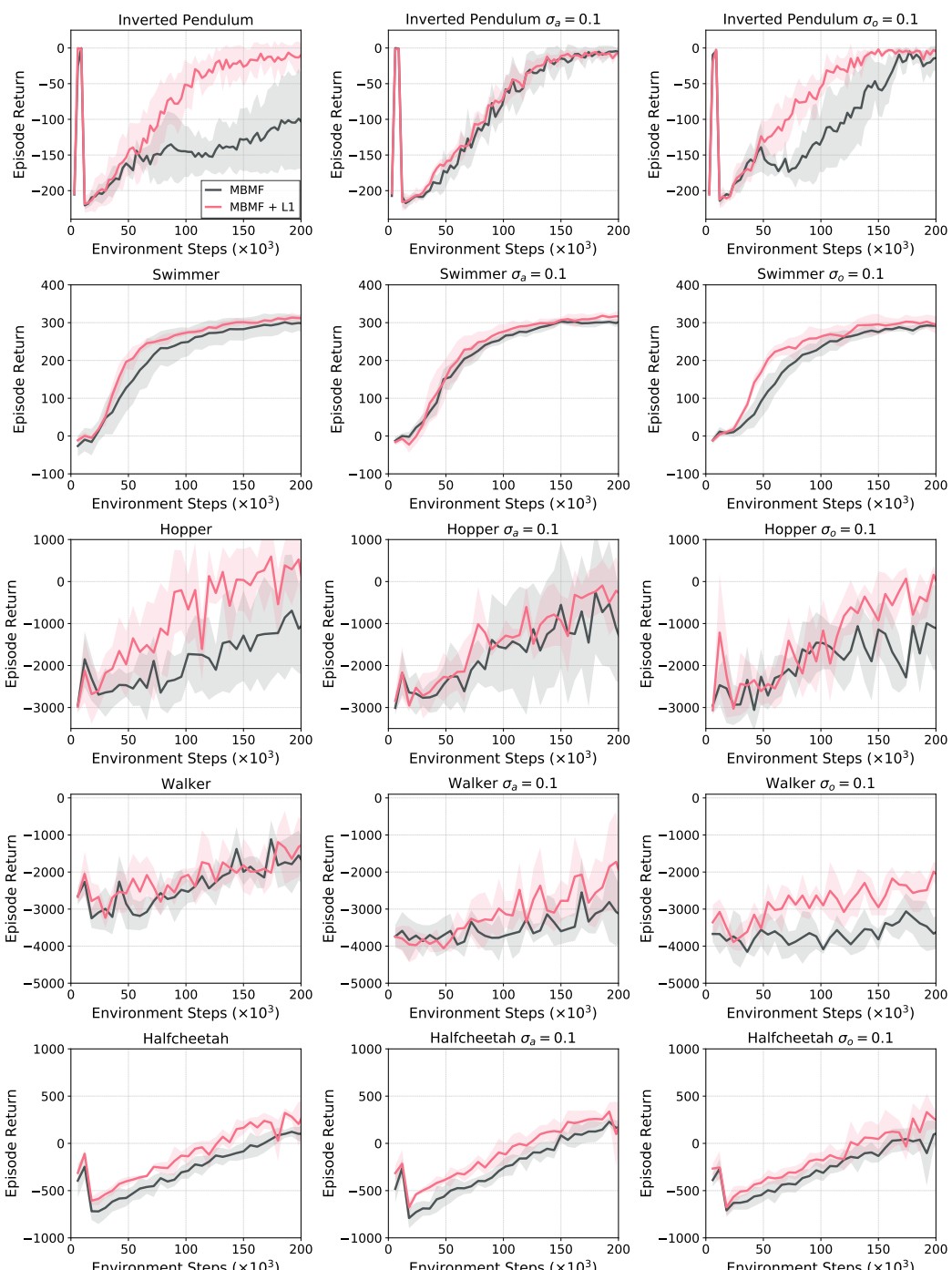

Figure 6: Plots of $\mathcal{L}_1$-MBMF learning curves as a function of episodic steps. The evaluation of the performance is identical to $\mathcal{L}_1$-METRPO.

### D.4 COMPARISON WITH PROBABILISTIC MODELS

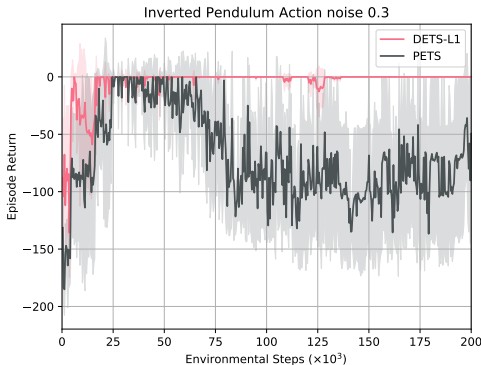

Figure 7: Plots of $\mathcal{L}_1$-DETS vs PETS learning curves as a function of episodic steps.

Probabilistic models, as discussed in (Chua et al., 2018; Wang & Ba, 2020), offer a common approach in Reinforcement Learning (RL) to tackle model uncertainty. In contrast, our approach, centered on a robust controller, shares a similar spirit but differs in architecture. While previous works directly integrate uncertainty into decision-making, for example, through methods like sampling-based Model Predictive Control (MPC) (Chua et al., 2018), our approach takes a unique path by decoupling the process. We address uncertainty by explicitly estimating and mitigating it based on the learned deterministic nominal dynamics, allowing the MBRL algorithm to operate as intended.

Recently, the authors in (Zheng et al., 2022) emphasized that the empirical success of probabilistic dynamic model ensembles is attributed to their Lipschitz-regularizing aspect on the value functions. This observation led to the hypothesis that the ensemble's key functionality is to regularize the Lipschitz constant of the value function, not in its probabilistic formulation. The authors have shown that the predictive quality of deterministic models does not show much difference with probabilistic (ensemble) models, leading to the conclusion that deterministic models can offer computational efficiency and practicality for many MBRL scenarios. In this context, our work exploits the practical advantages of using deterministic models, while $\mathcal{L}_1$ adaptive controller accounts for the randomness present in the environment.

In Fig. 7, we conducted supplementary experiments comparing PETS and its deterministic counterpart, DETS, with $\mathcal{L}_1$ augmentation. The results were obtained using multiple random seeds and 200,000 timesteps in the Inverted Pendulum environment with an action noise of $\sigma_a = 0.3$, demonstrating that the deterministic model with $\mathcal{L}_1$ augmentation ($\mathcal{L}_1$-DETS) can outperform the probabilistic model approach (PETS). However, it's important to note that this comparison is specific to one environment. We refrain from making broad claims regarding DETS's superiority over PETS without further in-depth analysis and experimentation. In conclusion, we express our intent to explore the development of $\mathcal{L}_1$-MBRL that can effectively work alongside probabilistic models, recognizing the potential advantages of both approaches.

