# OpenReview forum: "Robust Model Based Reinforcement Learning Using $\mathcal{L}_1$ Adaptive Control"
_ICLR.cc/2024/Conference — ICLR 2024 poster_

### Official Review · Reviewer_QEpj · 2023-10-26

**Soundness:** 3 good
**Presentation:** 4 excellent
**Contribution:** 3 good
**Rating:** 8
**Confidence:** 5

**Summary:**

The authors propose an $\mathcal{L}_1$ adaptive control "add-on" that can be applied to existing model-based RL algorithms in order to improve robustness to model uncertainties. This scheme is agnostic to the choice of model-based RL algorithm; it transforms/filters said algorithm's control outputs before they are input to the dynamical system or simulator. The authors provide some theory behind this approach as well as experimental validation in MuJoCo environments.

**Strengths:**

- The paper is written very well, which is rarely a given.
- The authors spend an appropriate amount of space on each section of the paper and do not belabor introductory material or time over-explaining things.
- The (meta-)algorithm is a nice idea that can be tested fairly easily.
- Code to replicate experiments was provided, which is rarely a given.

**Weaknesses:**

Some presentation issues:
- The single-column "wrapped" figures are awkward, especially on pg. 6.
- Tables (esp. numbers) should have larger fonts--shouldn't be below footnote size.
- In Table 1 and Fig. 3, what the bounds/error bars represent isn't stated.
#
- I would have liked to see on-off comparisons for more model-based RL algorithms as in Figure 3 (especially in the main body of the paper), as it's one of the paper's main selling points.
- While I appreciate the installation README, having one of the steps being "go get a MuJoCo license" isn't very user-friendly (not sure why it couldn't have been included?).

**Questions:**

As a sanity check, how does this perform on linear systems?

---

> ### Author Response · Authors · 2023-11-15
>
> We sincerely thank the reviewer for recognizing the novelty of the work and for appreciating the writing quality and the clarity of presentation on how the algorithm fits into the MBRL framework. We address below the reviewer’s questions and concerns raised, providing answers and additional explanations.
>
> > *Q1. As a sanity check, how does this perform on linear systems?*
>
> We are grateful for the reviewer's suggestion to test our method on a linear system. Please note that the inverted pendulum system used in our study can be considered a linear system in the vicinity of its equilibrium point. Consequently, we have extended our experiments to include the application of an LQR controller (designed based on the nominal dynamics) to the system, paralleling the three noise conditions (noise-free, $\sigma_a=0.1$, and $\sigma_o=0.1$) This comparison aims to contextualize the performance of our $\mathcal{L}_1$-MBRL approach with the nominal optimal controller. We chose $Q=I$ and $R=1$
>
> | Algorithm | noise-free | $\sigma_a=0.1$ | $\sigma_o=0.1$ |
> |---------|------|------|-------|
> | LQR | $0.0\pm 0.0$ |  $0.0\pm 0.0$ |  $-29.2\pm 24.1$ |
> | METRPO |  $-51.3\pm 67.8$ |  $-105.2\pm 81.6$ |  $-74.2\pm 74.5$ |
> | $\mathcal{L}_1$-METRPO|  $0.0\pm 0.0$ |  $0.0\pm 0.0$ |  $-21.3\pm 20.7$|
>
> The bottom two rows are obtained from Table 1 of our manuscript. We note that $\mathcal{L}_1$-METRPO matched the performance of LQR in all tested scenarios, whereas METRPO failed in all cases. This suggests that the $\mathcal{L}_1$ enhancement to METRPO can elevate the MBRL controller comparable to the capabilities of the nominal optimal controller in the presence of uncertainty.
>
> > *W1. The single-column "wrapped" figures are awkward, especially on pg. 6. 2. Tables (esp. numbers) should have larger fonts--shouldn't be below footnote size. 3. In Table 1 and Fig. 3, what the bounds/error bars represent isn't stated.1*
>
> We thank the reviewer for bringing this to our attention. We have edited the font size in the table to a footnote size and edited the figure/table captions in blue.
>
> > W2. *I would have liked to see on-off comparisons for more model-based RL algorithms as in Figure 3 (especially in the main body of the paper), as it's one of the paper's main selling points.*
>
> We concur with the reviewer's suggestion to perform additional ablation studies to compare the contributions of $\mathcal{L}_1$ adaptive control across various model-based algorithms. In our manuscript, we conducted over 180 simulation experiments, assessing the enhanced robustness in most scenarios. Notably, this potential improvement was observed in two types of MBRL algorithms; METRPO is a Dyna-style algorithm and MBMF belongs to the Shooting-style algorithm, representing the two major types of online MBRL algorithms among three [1]. We opted not to implement a policy-search-type algorithm due to their reliance on known dynamics. Due to the limitation in our computational resources, instead of running more simulations on similar algorithms that follow the same structure/style, we decided to move on to validating our framework with different baseline structures, for example, that of offline MBRL algorithms, which we aim to present in our future work.
>
> W3. *While I appreciate the installation README, having one of the steps being "go get a MuJoCo license" isn't very user-friendly (not sure why it couldn't have been included?).*
>
> We thank the reviewer for bringing this to our attention. We reflected on the reviewer’s suggestion and made necessary modifications to the README accordingly.
>
> [ References ]
>
> [1] Wang, Tingwu, Xuchan Bao, Ignasi Clavera, Jerrick Hoang, Yeming Wen, Eric Langlois, Shunshi Zhang, Guodong Zhang, Pieter Abbeel, and Jimmy Ba. "Benchmarking model-based reinforcement learning." arXiv preprint arXiv:1907.02057, 2019.

---

> > ### Comment · Reviewer_QEpj · 2023-11-15
> > **response to rebuttal**
> >
> > Thank you for reading and addressing the concerns I brought up.

---

### Official Review · Reviewer_26Jm · 2023-10-30

**Soundness:** 3 good
**Presentation:** 4 excellent
**Contribution:** 2 fair
**Rating:** 6
**Confidence:** 3

**Summary:**

The paper proposes robust control method combining existing MBRL algorithms and $\mathcal{L}_1$ adaptive control method. MBRL algorithm provides the reference control and $\mathcal{L}_1$ adaptive control is applied to reject the uncertainty.

The difficulty in combining MBRL and adpative control method is how to represent the model. The authors have chosen control-affine dynamics using Taylor approximation and switching law, and it seems to work well.

**Strengths:**

- The idea to use MBRL as reference control and then applying $\mathcal{L}_1$ adaptive control seems to straightforward and clear. The presentation is clear and well-organized.

- The method is applicable to wide range of MBRL algorithms.

- The experimental results show the effectiveness of the proposed method.

**Weaknesses:**

- Using output of the RL algorithm as reference input for $\mathcal{L}_1$ adaptive control has been explored in Cheng et al., therefore the proposed method does not seem to be novel.

- Since the suggested method is an augmentation framework, the performnace of the method highly depends on existing MBRL algorithms.

- Since the true model is assumed to be deterministic, it highly restricts the range of application.

- Comparison with existing uncertainty dealing methods seems to insufficient. I would expect several more explanations and experiments comparing with probabilistic models as in Section D.4.


Cheng, Yikun, et al. "Improving the Robustness of Reinforcement Learning Policies With ${\mathcal {L} _ {1}} $ Adaptive Control." IEEE Robotics and Automation Letters 7.3 (2022): 6574-6581.

**Questions:**

- What is the importance of Theorem 1 or what does it have to do with MBRL? What is the difference with the existing stability result in $\mathcal{L}_1$-adaptive control theory?

- In equation (9), how is $\nabla_u \hat{f}_{\theta}(x_t,u)$ implemented?

- What does it mean to skip the update in non-differentiable points? Then should the update is over when it the agorihm meets the non-differntiable point?

---

> ### Author Response · Authors · 2023-11-15
> **(Part 1/2) Answers to Weakness 1 and Questions**
>
> We thank the reviewer for acknowledging the clarity and effectiveness of our work. Below, we address any potential misunderstandings and subsequently respond to the raised questions.
>
> > *W1. Using output of the RL algorithm as reference input for $\mathcal{L}_1$ adaptive control has been explored in Cheng et al., therefore the proposed method does not seem to be novel.*
>
> We highlight a crucial distinction between Cheng et al. [1] and our work: 1) we do not presume knowledge of the nominal control-affine dynamics, 2) and we do not assume the availability of a pre-trained (well-performing) policy. In [1], the policy is initially learned in an offline setting (in a model-free sense), where sufficient data—both in quantity and quality—can be collected from a simulator. Subsequently, when implementing this policy in a real environment with uncertainty, $\mathcal{L}_1$ adaptive control (designed based on nominal dynamics) is employed. (For further discussion, please refer to our rebuttal to Reviewer NiYe)
>
> In contrast, our work establishes a framework for utilizing adaptive control when neither the nominal model nor a comprehensive simulator is available. In our scenario, we iteratively update the model, which serves as the simulator to train the policy and use this policy to collect data from a noisy environment. This problem setup requires many additional considerations (e.g. treating nonlinear models, designing switching conditions, etc) to implement a robust controller. Providing novel solutions to handle these intricacies, our work significantly diverges from the problem and solution proposed in [1]. In addition, our framework for implementing an $\mathcal{L}_1$ adaptive controller in the training loop has not been presented elsewhere.
>
> > *Q1. What is the importance of Theorem 1 or what does it have to do with MBRL? What is the difference with the existing stability result in  $\mathcal{L}_1$ -adaptive control theory?*
>
> The Theorem 1 serves to quantify the predictive quality of the state predictor in our add-on scheme concerning model approximation errors $\epsilon_l$ and $\epsilon_a$, and parameters governing the adaptation law ($T_s$, $\lambda_{\min}$). It confirms that our state predictor assures a bounded prediction error, which is important in our design where the performance is inherently tied to the predictive quality of the uncertainty estimate. On the other hand, the results in $\mathcal{L}_1$ adaptive control theory encompass the stability of the system in the closed loop. The discrepancy in the objectives of the two analyses is mainly attributed to the availability of the nominal dynamics (i.e. stability, equilibrium, etc) assumed in $\mathcal{L}_1$ adaptive control problems.
>
> > *Q2. In equation (9),  how is $\nabla \hat{f}_{\theta}(x_t,u_t)$ implemented?*
>
> Differentiation of the neural network (NN) model occurs during the backpropagation process used for updating the model. In this procedure, we write the Jacobian of the NN model, denoted as $\nabla_u \hat{f}_{\theta}(x,u)$, as a function and make function calls whenever necessary. An alternative approach is to compute the Jacobian of the NN using Autograd, a technique that numerically evaluates the Jacobian and is commonly referred to as automatic differentiation [3]
>
> > *Q3. What does it mean to skip the update in non-differentiable points? Then should the update is over when it the agorihm meets the non-differntiable point?*
>
> This is a technical statement to circumvent the nondifferentiability of an NN (e.g. ReLU at zero), at which our affinized dynamics (Eq.(9) in our manuscript) is not defined. The statement clarifies that we apply exceptions to the switching law (10) specifically at those points. In practice, replacing a nondifferentiable activation function with its differentiable counterpart (Swish for ReLU) can easily resolve these nondifferentiability concerns.

---

> ### Author Response · Authors · 2023-11-15
> **(Part 2/2) Comments on Weakness**
>
> > *W2. Since the suggested method is an augmentation framework, the performance of the method highly depends on existing MBRL algorithms.*
>
> We agree with the reviewer that the suggested framework is constrained by the performance of the current MBRL algorithm, as highlighted in the limitation section of our work. The primary motivation for our proposed framework arises from the fact that introducing noise can have adverse effects on the performance of MBRL algorithms [4], which in turn may limit their applicability to real-world systems. We contend that our proposed methodology holds the potential to narrow the performance gap between algorithms with access to clean datasets and those without.
>
> > *W3. Since the true model is assumed to be deterministic, it highly restricts the range of application. Comparison with existing uncertainty dealing methods seems to be insufficient. I would expect several more explanations and experiments comparing with probabilistic models as in Section D.4.*
>
> We acknowledge the reviewer's concern that the deterministic dynamics assumption limits the range of application. However, we would like to emphasize that as mentioned in Section D.4 of the Appendix, our approach to using a robust controller is in a similar spirit to that of the probabilistic models but simply differs in architecture to address the same issue. Our approach uses an explicit method to estimate and reject uncertainty based on the deterministic nominal (learned) dynamics, while the probabilistic models use the variance of the prediction to handle uncertainty.
>
> Recently, the authors in [2] emphasized that the empirical success of probabilistic dynamics model ensembles is attributed to their Lipschitz-regularizing aspect on the value functions. This observation led to the hypothesis that the ensemble's key functionality is to regularize the Lipschitz constant of the value function, not in the probabilistic formulation itself. The authors have shown that the predictive quality of deterministic models does not show much difference with probabilistic (ensemble) models, leading to the conclusion that deterministic models can offer computational efficiency and practicality for many MBRL scenarios. In this context, our work exploits the practical advantages of using deterministic models, while $\mathcal{L}_1$ adaptive controller accounts for the randomness present in the environment.
>
> The above explanation is added to Section D.4 for the clarity of the manuscript, as suggested by the reviewer.
>
> [ References ]
>
> [1] Cheng, Yikun, Pan Zhao, Fanxin Wang, Daniel J. Block, and Naira Hovakimyan. "Improving the Robustness of Reinforcement Learning Policies With ${\mathcal {L} _ {1}} $ Adaptive Control." IEEE Robotics and Automation Letters 7, no. 3 (2022): 6574-6581.
>
> [2] Zheng, Ruijie, Xiyao Wang, Huazhe Xu, and Furong Huang. "Is Model Ensemble Necessary? Model-based RL via a Single Model with Lipschitz Regularized Value Function." In The Eleventh International Conference on Learning Representations. 2022.
>
> [3]  Atilim Gunes Baydin, Barak A. Pearlmutter, Alexey Andreyevich Radul, Jeffrey Mark Siskind. Automatic differentiation in machine learning: a survey. The Journal of Machine Learning Research, 18(153):1–43, 2018
>
> [4] Wang, Tingwu, Xuchan Bao, Ignasi Clavera, Jerrick Hoang, Yeming Wen, Eric Langlois, Shunshi Zhang, Guodong Zhang, Pieter Abbeel, and Jimmy Ba. "Benchmarking model-based reinforcement learning." arXiv preprint arXiv:1907.02057, 2019.

---

> > ### Comment · Reviewer_26Jm · 2023-11-16
> > **Response to authros**
> >
> > Thank you for the detailed response and most of the concerns have been addressed. I have increased my score from 5 to 6.

---

### Official Review · Reviewer_QoUP · 2023-10-30

**Soundness:** 3 good
**Presentation:** 3 good
**Contribution:** 3 good
**Rating:** 6
**Confidence:** 2

**Summary:**

This paper presents an improvement to existing model based RL algorithms where an approximate dynamics model is used in an L1 adaptive control scheme to contribute to the robustness of the algorithms. By relying on an "affinized" a trained nonlinear model using the switching rule, L1 adaptive control can be used in tandem with MBRL algorithms. This method is shown to improve both the sample efficiency and robustness of MBRL algorithms.

**Strengths:**

1. The statement of contributions is nice for understanding which parts of the proposed work are original and which already exist.
2. There is a nice literature review in the related work and preliminaries sections to introduce L1 adaptive control to people less familiar with it.
3. The approach was shown to work well on a wide variety of control tasks.

**Weaknesses:**

1. Without a solid adaptive control background, I found this paper very hard to follow.
2. Some of the figure captions weren't descriptive enough, more detail would be helpful.
3. This paper seemed to be very heavy on math in place it made it hard to follow.

**Questions:**

Why L1 adaptive control? What were some of the other options from the adaptive control community considered in this work?

---

> ### Author Response · Authors · 2023-11-15
>
> We thank the reviewer for acknowledging the novelty of our work and for appreciating the clarity of presentation and effectiveness of the proposed framework. We first answer to the reviewer’s question, and some additional comments on the Weaknesses of our work are followed.
>
> > *Q. Why $\mathcal{L}_1$  adaptive control? What were some of the other options from the adaptive control community considered in this work?*
>
> $\mathcal{L}_1$ adaptive control’s use is particularly appealing to us because because it does not need any assumptions on the structure of the uncertainties which are almost always unverifiable.
>
> To provide the reviewer with a recent work that applied a different approach, [1] used a model-reference adaptive control (MRAC) scheme to handle parametric uncertainties. While this approach has a distinct theoretical objective based on different assumptions, this method requires restrictive assumptions to know the nominal dynamics and a well-trained policy. In contrast, $\mathcal{L}_1$ adaptive control imposes fewer assumptions, enhancing its usability in an MBRL setup.
>
> > *W1. Without a solid adaptive control background, I found this paper very hard to follow.*
>
> > *W3. This paper seemed to be very heavy on math in place it made it hard to follow.*
>
> We agree with the reviewer’s assessment that the paper will appear unapproachable for readers without a background in adaptive control. Due to space limitations and to avoid overburdening the reader, Section 2.2 on $\mathcal{L}_1$ in the manuscript is intentionally kept succinct. However, for readers seeking a deeper understanding, we have provided a detailed explanation of $\mathcal{L}_1$ adaptive control in Section A of the Appendix. For an alternative explanation with a physical example, we refer the reviewer to [2].
>
> > *W2. Some of the figure captions weren't descriptive enough, more detail would be helpful.*
>
> We thank the reviewer for pointing this out. The manuscript has been edited in blue, and we would like to know if the modifications are helpful.
>
> [ References ]
>
> [1] Annaswamy, A. M., Guha, A., Cui, Y., Tang, S., Fisher, P. A., & Gaudio, J. E. (2023). Integration of adaptive control and reinforcement learning for real-time control and learning. IEEE Transactions on Automatic Control.
>
> [2] Wu, Zhuohuan, Sheng Cheng, Kasey A. Ackerman, Aditya Gahlawat, Arun Lakshmanan, Pan Zhao, and Naira Hovakimyan. "$\mathcal{L}_1$ adaptive augmentation for geometric tracking control of quadrotors." In 2022 International Conference on Robotics and Automation (ICRA), pp. 1329-1336. IEEE, 2022.

---

### Official Review · Reviewer_NiYe · 2023-10-31

**Soundness:** 2 fair
**Presentation:** 2 fair
**Contribution:** 3 good
**Rating:** 6
**Confidence:** 4

**Summary:**

This paper proposes a control-theoretic augmentation scheme for Model-Based Reinforcement Learning (MBRL) algorithms. This method is designed to enhance the robustness of the MBRL system against uncertainties, using MBRL perturbed by the L1 adaptive control.

**Strengths:**

The paper attempts to balance theory and empirical validation. It aims to integrate Model-Based Reinforcement Learning with control.

**Weaknesses:**

The paper has flaws both with theory and experiment.

  1. In theory, there is so much technically that is incorrect.  Some issues are highlighted below.

  2. The evaluation does not compare the proposed method with recent competitors, such as:

Annaswamy, A. M., Guha, A., Cui, Y., Tang, S., Fisher, P. A., & Gaudio, J. E. (2023). Integration of adaptive control and reinforcement learning for real-time control and learning. IEEE Transactions on Automatic Control.

Kim, J. W., Park, B. J., Yoo, H., Oh, T. H., Lee, J. H., & Lee, J. M. (2020). A model-based deep reinforcement learning method applied to finite-horizon optimal control of nonlinear control-affine system. Journal of Process Control, 87, 166-178.

The paper is littered with inconsistencies and half-truths that make it difficult to follow.

A. The paper has serious misunderstandings concerning the relationships among non-linear or affine models and switching. The paper makes claims that mis-represent the following:

1. In control theory, using a switching law is independent of whether the system is non-linear or affine.
2. An affine model can be tuned to be a close approximation to a non-linear model.
3. A well-tuned affine model can be compatible with the underlying MBRL algorithm.

The text around eq. 10-11 is standard control theory: we tune an affine model for a set of operating conditions, and have a switching function that switches to the best-approximation affine model as appropriate.

It is confusing to try to follow the logic of p. 5, starting with "Although using the above naive control-affine model can be convenient, it must trade in the capabilities of the underlying MBRL algorithm."

There are many overblown claims. such as:

  (i) "Classical control tools rely on extensively modeled dynamics that are gain scheduled, linear, and/or true up to parametric uncertainties. An example is prohibitively expensive wind-tunnel modeling for designing flight control systems (Neal et al., 2004; Nichols et al., 1993)."

This is not true. Many modern approaches use nonlinear methods. Your example makes no sense.

  (ii) "MBRL algorithms often use highly nonlinear models (often NNs) that do not have true parameters corresponding to the ground truth dynamics, only optimal from a predictive sense, which makes consolidating MBRL with control theoretic tools challenging."

The following is over 10 years old and covers many topics in (ii):

Wang, X., & Hovakimyan, N. (2012). L1 adaptive controller for nonlinear time-varying reference systems. Systems & Control Letters, 61(4), 455-463.


p.4: "it is necessary to represent the nominal model in the control-affine form."

It is not required to have affine form, even for NNs:

Padhi, R., Unnikrishnan, N., & Balakrishnan, S. N. (2007). Model-following neuro-adaptive control design for non-square, non-affine nonlinear systems. IET Control Theory & Applications, 1(6), 1650-1661.

**Questions:**

1. What is the extension over:

Annaswamy, A. M., Guha, A., Cui, Y., Tang, S., Fisher, P. A., & Gaudio, J. E. (2023). Integration of adaptive control and reinforcement learning for real-time control and learning. IEEE Transactions on Automatic Control.

2. Please carefully show differences with the following paper. There also seems to be a big overlap with:

Kim, J. W., Park, B. J., Yoo, H., Oh, T. H., Lee, J. H., & Lee, J. M. (2020). A model-based deep reinforcement learning method applied to finite-horizon optimal control of nonlinear control-affine system. Journal of Process Control, 87, 166-178.

---

> ### Author Response · Authors · 2023-11-15
> **(Part 1/2) Answers to Questions**
>
> We thank the reviewer for providing insightful comments. We first provide answers to their [Questions], and subsequently address some of the points raised under the [Weakness] section of their review.
>
> > *Q1. What is the extension over [1]?*
>
> While our work and [1] both integrate RL policy with adaptive control, our work significantly diverges in key aspects such as the underlying assumptions, the scope of the study, and the usability of the method.
>
> **[Availability of the nominal model and well-trained policy]**  The major difference between Annaswamy et.al [1] and our work is that we do not have access to the nominal dynamics and a well-trained policy. In [1], the policy is assumed to be acquired from an offline setting where sufficient data (both in quality and quantity) can be collected. A well-trained policy is meant to locally stabilize the uncertain system and ensure its uniform boundedness. Then the policy is employed together with an adaptive control (designed based on the nominal dynamics) in the closed loop. In contrast, our work develops a framework to incorporate a robust controller within an MBRL architecture wherein both the model and policy undergo iterative updates. As we state in the paper, one of the main guiding principles of our work is to place no additional burden or requirements on the high-level RL control, whereas, the work in [1] makes explicit requirements from the nominal dynamics and RL policy.
>
> **[Affine approximation and Switching conditions]** The reviewer implied in [Weakness] that we can tune a piecewise-affine (PWA) system to approximate the NN model, and then take a similar approach from [1]. However, due to the frequent updates of an NN model in an MBRL algorithm, such an approach becomes infeasible or impractical in an MBRL context. That is, optimizing for both the number of affine submodels and the parameters of each submodel is a well-known NP-hard problem. Even if we heuristically fix the number of submodels, applying a PWA system identification at each iteration will significantly increase the computational burden of the algorithm, and importantly, the accuracy of the affine approximation is not controlled. We concluded this approach is neither scalable nor generalizable, at least in the online MBRL setting that our work is focusing on.
>
> For this reason, many works follow the simpler approach of restricting the NN model structure to the control-affine class [2-3]. However, as evident in Figure 2 of our manuscript, this strategy can significantly compromise the performance of the baseline algorithm. To address this issue, we propose to obtain affine dynamics according to a state-dependent switching law, i.e., we do not predefine sub-models, but instead generate affine models along the trajectory. This methodology allows us to explicitly constrain and control the affinization error, a critical factor for providing predictive guarantees for the $\mathcal{L}_1$ adaptive control (Theorem 1 of our manuscript). Moreover, the Jacobian evaluation of a model at the switching point can reuse the stored information from the backpropagation during the model training. Thus, the computational burden of the algorithm is significantly less compromised compared to performing PWA system identification at each iteration.
>
> **[Parametric uncertainty]** The authors in [1] only considered linearly parameterizable uncertainty (Assumption 3, [1]) and its extension to bounded (non-state dependent) perturbations (Assumption 5, [1]). While this assumption is made to provide theoretical guarantees, it lacks the accuracy to capture reality. In our case, we place no such restrictive assumptions.
>
> > *Q2. Please carefully show differences with the following paper. There also seems to be a big overlap with [4]*
>
> While [4] shares some similar ideas with our work, the assumptions and methods to solve an optimal control problem are largely different from our work. This work used the term ‘model-based’ RL to represent the iterative learning process for the value, costate, and policy functions (Eq. (25), [4]) using the globalized dual heuristic programming (GDHP) algorithm when the dynamical model is assumed to be known and the objective function is quadratic. Their contribution is to enhance the GDHP update rule which relies on a noisy state measurement by formulating DNN approximations for each value, costate, and policy function to generate ‘targets’ (Equation (26)-(28), [4]), and their corresponding loss functions (Equation (32)-(34), [4]).
>
> Our method does not assume the availability of the nominal model or a quadratic objective function, and we do not model value and costate functions to develop a policy. In addition, although they use the term ‘model-based’ approach, they do not learn the dynamical model, and therefore, our work and theirs share a similarity only in name.

---

> > ### Comment · Reviewer_NiYe · 2023-11-22
> > **Response to author rebuttal**
> >
> > Thanks to the authors for the clarifications. If the authors can focus their contributions more precisely in line with this review (arguments agreed by the authors) then I can change my assessment to above the threshold for publication.  The original submission needs revision in many sections, and the contributions must be sharpened to avoid the false claims of the original version.

---

> > > ### Author Response · Authors · 2023-11-22
> > >
> > > We appreciate the reviewer's re-evaluation of our paper upon further revisions of the sentences identified in their review. We further revised these sentences as follows for better sharpness, and they are also revised in line with our manuscript.
> > >
> > > >Previous: Classical control tools often rely on extensively modeled dynamics that are gain scheduled, linear, control affine, and/or true up to parametric uncertainties.
> > >
> > > Revised: In order to analyze systems and design controllers for such systems, conventional control methods often assume extensively modeled dynamics that are gain scheduled, linear, control affine, and/or true up to parametric uncertainties.
> > >
> > > >Previous: MBRL algorithms frequently update highly nonlinear models (e.g. NNs) to enhance their predictive accuracy. This iterative process and the high nonlinearity of the model present a challenge in the integration of robust and adaptive controllers into MBRL algorithms.
> > >
> > > Fixed: MBRL algorithms frequently update highly nonlinear models (e.g. NNs) to enhance their predictive accuracy. The combination of this iterative updating and the model’s high nonlinearity creates a unique challenge in embedding robust and adaptive controllers within MBRL algorithms.

---

> ### Author Response · Authors · 2023-11-15
> **(Part 2/2) Comments on Weakness**
>
> Comments regarding the affinization and switching law are discussed in our first comment. Here, we discuss the remaining comments made in the [Weakness] section of the review.
>
> > *W1. There are many overblown claims. such as (i). This is not true. Many modern approaches use nonlinear methods. Your example makes no sense.*
>
> We agree with the reviewer and we acknowledge that our statement could be misinterpreted as an absence of nonlinear approaches in control theory. While control of nonlinear systems has an extensive history, using RL with adaptive control for unknown nonlinear systems is new. Indeed, the reviewer provided a very new work [1] (published in 2023), and in our answer to Q1 above, we provided ample evidence of simplifying assumptions used in [1] for the applicability of adaptive control. This highlights that while many modern approaches use nonlinear methods, we concern ourselves with one particular type of it. We revised this sentence in the edited version (p.1 in blue) of our manuscript as follows:
>
> *Classical control tools often rely on extensively modeled dynamics that are gain scheduled, linear, control affine, and/or true up to parametric uncertainties.*
>
> > *W2. The following is over 10 years old and covers many topics in (ii) [5].*
>
> We agree with the reviewer that many works including [5] provide robust adaptive control strategies for nonlinear systems. We have edited this sentence to clarify our intent as follows (p.1-2 in our manuscript):
>
> *MBRL algorithms frequently update highly nonlinear models (e.g. NNs) to enhance their predictive accuracy. This iterative process and the high nonlinearity of the model present a challenge in the integration of robust and adaptive controllers into MBRL algorithms.*
>
> While we revised the sentence for clarity, we point out that in [5], authors assume the knowledge of the nonlinear control-affine time-varying system ($f$ and $g$ in [5]). Furthermore, a Lyapunov function for the nonlinear system is assumed to exist. Exploiting these assumptions, they design a control strategy to compensate for the unmodeled uncertainty $h$. However, we do not place the assumption on the knowledge of a stabilizable system (with Lyapunov certificates) which makes our problem statement significantly different from [5].
>
> > *W3. it is necessary to represent the nominal model in the control-affine form."  It is not required to have an affine form, even for NNs [6]*
>
> In [6], the authors represent the uncertainties with an NN and update its weights according to Eqn. (44) in [6]. However, the NN used (see Sec. 2.3.1-2.3.2) is represented so that each element of the output can be represented linearly on the network weights. The only two cases where this can happen are if i) the neural network only has a single layer, or ii) one assumes that except the output layer, all other layers’ weights are correct (in some sense). This assumption therefore leads to the same assumption as in [1] of linearly parametrized uncertainties and thus the authors of [6] can apply the standard gradient flow adaptation law. We place no such restrictive assumptions, and we do not assume any knowledge of the system dynamics. Our controller knows only what the high-level MBRL algorithm knows: a purely data-driven NN model of the dynamics.
>
> Hence, while the reviewer’s claim that the use of NNs does not require affine forms is technically correct, the example provided in [6] hides many restrictive and simplifying assumptions using which one can easily apply any flavor of adaptive control. Unfortunately, such assumptions are unverifiable for complex systems of the type that MBRL is capable of controlling and the examples we provide in the manuscript.
>
> [1] Annaswamy, A. M., Guha, A., Cui, Y., Tang, S., Fisher, P. A., & Gaudio, J. E. (2023). Integration of adaptive control and reinforcement learning for real-time control and learning. IEEE Transactions on Automatic Control.
>
> [2] Xiong, Zhihua, and Jie Zhang. "Modelling and optimal control of fed-batch processes using a novel control affine feedforward neural network." Neurocomputing 61 (2004): 317-337.
>
> [3] Gil, Paulo, Jorge Henriques, Alberto Cardoso, Paulo Carvalho, and António Dourado. "Affine neural network-based predictive control applied to a distributed solar collector field." IEEE Transactions on control systems technology 22, no. 2 (2013): 585-596.
>
> [4] Kim, J. W., Park, B. J., Yoo, H., Oh, T. H., Lee, J. H., & Lee, J. M. (2020). A model-based deep reinforcement learning method applied to finite-horizon optimal control of nonlinear control-affine system. Journal of Process Control, 87, 166-178.
>
> [5] Wang, X., & Hovakimyan, N. (2012). L1 adaptive controller for nonlinear time-varying reference systems. Systems & Control Letters, 61(4), 455-463.
>
> [6] Padhi, R., Unnikrishnan, N., & Balakrishnan, S. N. (2007). Model-following neuro-adaptive control design for non-square, non-affine nonlinear systems. IET Control Theory & Applications, 1(6), 1650-1661.

---

> ### Author Response · Authors · 2023-11-20
> **Rebuttal reminder**
>
> Dear Reviewer NiYe
>
> We have carefully addressed your comments in our author response, highlighting key differences from Annaswamy et al. [1] and Kim et al. [4]. We would appreciate your feedback. If there are any further concerns, we are committed to addressing them during the discussion period. If the reviewer’s concerns are addressed sufficiently, we would like to sincerely request a re-evaluation of our manuscript.
>
> Best Regards,
>
> Authors.

---

### Author Response · Authors · 2023-11-23

Dear Reviewers,

We sincerely appreciate your dedication and valuable insights during the review process. During the discussion period, we identified certain sentences and captions that required revisions for enhanced clarity. These have been addressed in our manuscript, initially highlighted in blue for easy identification by the reviewers. As we approached the end of the discussion period, we finalized our manuscript, converting all the blue text back to black.

Best regards,

The Authors.

---

### Meta-Review · Area_Chair_DCcw · 2023-12-06

**Metareview:**

*Summary*: This paper presents an L1-adaptive-control-based augmentation method for MBRL algorithms, to improve the robustness and efficiency of existing MBRL algorithms. The key idea is to learn control-affine dynamics models with a switching law and use L1 adaptive control to perturb the nominal output of any MBRL algorithm. Multiple MuJoCo experiments have been conducted to evaluate the effectiveness of the proposed augmentation methods on top of various MBRL algorithms.

*Strengths*: (1) Well-motivated (existing MBRL algorithms do suffer from robustness issues). (2) Clean algorithm design with theoretical justification.

*Weaknesses*: (1) Need more justification of why use the particular L1 adaptive control method for augmentation. (2) Hard to follow for people not familiar with adaptive control theory (e.g., MBRL community). (3) No real-world experiments.

**Justification For Why Not Higher Score:**

See the weakness part.

**Justification For Why Not Lower Score:**

See the strength part.

---

### Decision · Program_Chairs · 2024-01-16

Accept (poster)